# Exploring Unified Perspective For Fast Shapley Value Estimation

## Abstract

Shapley values have emerged as a widely accepted and trustworthy tool, grounded in theoretical axioms, for addressing challenges posed by black-box models like deep neural networks. However, computing Shapley values encounters exponential complexity in the number of features. Various approaches, including ApproSemivalue, KernelSHAP, and FastSHAP, have been explored to expedite the computation. We analyze the consistency of existing works and conclude that stochastic estimators can be unified as the linear transformation of importance sampling of feature subsets. Based on this, we investigate the possibility of designing simple amortized estimators and propose a straightforward and efficient one, **SimSHAP**, by eliminating redundant techniques. Extensive experiments conducted on tabular and image datasets validate the effectiveness of our SimSHAP, which significantly accelerates the computation of accurate Shapley values.

## 1 Introduction

Deep learning techniques have made significant contributions across numerous industries due to the remarkable ability to learn complex functions both efficiently and accurately. However, lack of interpretability hinders further application of black-box models (e.g., deep neural networks) in trust-demanding areas such as autonomous driving and healthcare. Explanability aims to establish a stable mapping from abstract representation space to understandable human concept space (Gilpin et al., 2018; Zhang et al., 2021). Grounded on four fairness-based axioms (efficiency, symmetry, linearity, and dummy player), Shapley values provide a stable and unique linear additive explanation by computing the linear combination of marginal contributions of each feature based on cooperative game theory (Shapley et al., 1953). However, the computation cost of Shapley values becomes prohibitive for high-dimensional data (Van den Broeck et al., 2022).

To address this high complexity, various approaches have been proposed to accelerate the computation of Shapley values, which can be categorized into model-agnostic and model-specific methods (Chen et al., 2023a). **Model-agnostic methods**, such as semivalue (Castro et al., 2009) and least squares value (Lundberg & Lee, 2017; Covert & Lee, 2021), approximate Shapley values by sampling subsets of feature combinations. **Model-specific methods** aim to introduce model-specific information for accelerating estimation. For simple linear models, the computation cost can be reduced from exponential to linear (Lundberg & Lee, 2017). Attempts have also been made to accelerate Shapley value computation for tree-based models (Lundberg et al., 2020) and neural networks (Wang et al., 2021; Jethani et al., 2021; Chen et al., 2023b). Although there has been rapid

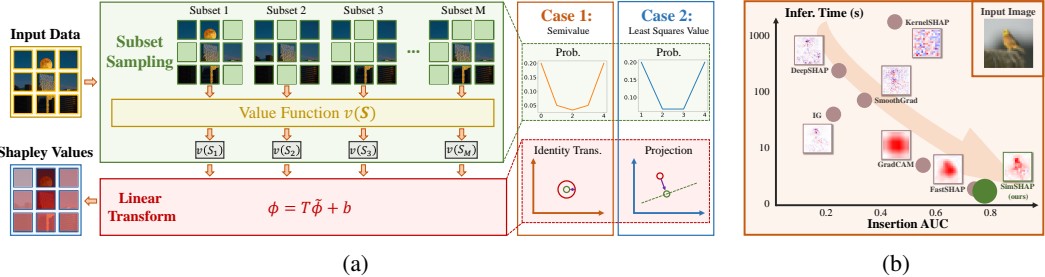

(a)                                                                 (b)

Figure 1: (a) Existing stochastic estimators for Shapley values can be unified as a linear transformation of the values obtained from sampled subsets. (b) We propose SimSHAP, which achieves high efficiency and maintain competitive approximation accuracy.

progress in the research of Shapley value acceleration, the exact differences among these algorithms remain unclear, making it challenging to select the appropriate algorithm for real-world scenarios. Therefore, it is necessary to explore the relationship among these algorithms.

This paper focuses on examining estimation strategies for Shapley values, including semivalue (Castro et al., 2009) and least squares value (Lundberg & Lee, 2017; Covert & Lee, 2021). It is observed that these strategies exhibit no substantial differences, and then we propose a unified perspective on stochastic estimators, which can be regarded as the linear transformation of the values obtained from sampled subsets (see Fig. 1a). Furthermore, recent amortized estimators can also be unified as a fitting problem to Shapley values within different metric spaces. Motivated by the principle of simplicity, we propose **SimSHAP**, a simple and fast amortized Shapley value estimator. SimSHAP trains an amortized explanation model by minimizing the $l2$-distance to the estimated Shapley values within the Euclidean space. Compared to conventional methods (Castro et al., 2009; Lundberg & Lee, 2017), SimSHAP achieves orders of magnitude faster computation with comparable accuracy as shown in Fig. 1b. Additionally, compared to recent amortized methods (Jethani et al., 2021), SimSHAP adopts an unconstrained optimization approach without subtle normalization (Ruiz et al., 1998). Extensive experiments on tabular and image datasets have been conducted to demonstrate the effectiveness of SimSHAP. Our contributions are summarized as follows:

- **Unified perspective of Shapley value estimation.** We unify various Shapley value estimation strategies as the linear transformation of values obtained from sampled subsets.
- **SimSHAP, a simple and fast amortized estimator.** We propose SimSHAP, which minimizes the $l2$-distance to the approximated Shapley values within the Euclidean space.
- **Consistent efficiency improvement.** We show consistent efficiency improvement while maintaining accuracy through extensive experiments on both tabular and image datasets.

## 2 METHOD

In this section, we first define the notations adopted throughout the paper and the preliminary knowledge of Shapley values in Section 2.1. Subsequently, we introduce the unified perspective of Shapley value estimation especially for semivalue and least squares value in Sections 2.2 and 2.3. Finally, we propose SimSHAP, a simple and fast amortized Shapley value estimator in Section 2.4.

### 2.1 SHAPLEY VALUES

For a typical classification model $f : \mathcal{X} \mapsto \mathcal{Y}$, we denote the input vector as $\boldsymbol{x} = (x_1, \cdots, x_d) \in \mathcal{X}$ with $d$ dimensions and $y \in \mathcal{Y} = \{1, \cdots, K\}$ be the corresponding label. Let $\boldsymbol{S} \subset \boldsymbol{N} = \{1, \cdots, d\}$ be subsets of all feature indices and $\mathbf{1}^{\boldsymbol{S}} \in \{0, 1\}^d$ be the corresponding indicator vector, where $\mathbf{1}_i^{\boldsymbol{S}} = 1$ if $i \in \boldsymbol{S}$ and 0 otherwise. Shapley values are first introduced in cooperative game theory (Shapley et al., 1953) for fair profit allocation among players, which are formulated as follows:

**Definition 1.** *(Shapley Values) For any value function $v : P(\boldsymbol{N}) \mapsto \mathbb{R}$, where $P(\boldsymbol{N})$ is the power set of $\boldsymbol{N}$, the Shapley values $\phi(v) \in \mathbb{R}^d$ is computed by averaging the marginal contribution of each feature over all possible feature combinations as:*

$$\phi_i = \sum_{\boldsymbol{S} \subset \boldsymbol{N} \setminus \{i\}} \frac{|\boldsymbol{S}|!(d - |\boldsymbol{S}| - 1)!}{d!} \left( v(\boldsymbol{S} \cup \{i\}) - v(\boldsymbol{S}) \right), \ \ i = 1, \cdots, d. \tag{1}$$

**Remark.** *Shapley values are the unique linear additive explanation that satisfies four fairness-based axioms (efficiency, symmetry, linearity, and dummy player) (Shapley et al., 1953).*

However, there are two considerations when applying Shapley values: (1) Machine learning models are not cooperative games; and (2) the complexity of Eq. (1) grows exponentially with the data dimension $d$. Value function choices and estimation strategies are two key factors for applying Shapley values to machine learning models. Numerous research investigate the value function of Shapley values for machine learning models (Frye et al., 2021; Covert et al., 2021; Aas et al., 2021), which describe the the meaning of absence and presence of each feature. In this work, we adopt the classic value function $v(\boldsymbol{S}) = f(\mathbf{1}^{\boldsymbol{S}} \odot \boldsymbol{x})$ with masked input corresponding to the feature indices $\boldsymbol{S}$. We recommend readers to refer to Chen et al. (2023a) for a comprehensive review of value function settings, and we mainly focus on estimation strategies.

### 2.2 ESTIMATION STRATEGY

To accelerate the computation of Shapley values, an intuitive idea is to approximate Eq. (1) by stochastic estimator such as Monte Carlo sampling. If considering the specific characteristics of ma-

chine learning models, we can further reduce the computation cost by merging or pruning redundant feature combinations. In this section, we first introduce three widely-used stochastic estimators and one recent amortized estimator, and then discuss about the inner relationship among them.

### 2.2.1 STOCHASTIC ESTIMATOR

**Semivalue** In previous work, the classic formulation of Shapley values as Eq. (1) is named as **semivalue** (Dubey et al., 1981). The marginal contribution $v(\boldsymbol{S} \cup \{i\}) - v(\boldsymbol{S})$ represents the contribution of feature $i$ cooperated with subset $\boldsymbol{S}$. Therefore, previous works, such as ApproSemivalue (Castro et al., 2009), propose to estimate Shapley values by importance sampling as follows:

$$\phi_i^{sv}(v) = \mathop{\mathbb{E}}_{\boldsymbol{S} \sim p^{sv}(\boldsymbol{S})} [v(\boldsymbol{S} \cup \{i\}) - v(\boldsymbol{S})] \approx \frac{1}{M} \sum_{k=1}^{M} [v(\boldsymbol{S}_k \cup \{i\}) - v(\boldsymbol{S}_k)], \quad (2)$$

where $p^{sv}(\boldsymbol{S}) = (|\boldsymbol{S}|!(d - |\boldsymbol{S}| - 1)!)/d!$ is importance sampling distribution and $\boldsymbol{S}_k \sim p^{sv}(\boldsymbol{S})$ is the $k$-th sampled subset. If the value function $v(\cdot)$ is bounded, by Lindeberg-Levy Central Limit Theorem, the variance of the random variable is related to the quantity M and is of order $\mathcal{O}(1/\sqrt{M})$.

**Random Order Value** Semivalue has an equivalent formulation named as **random order value** (Shapley et al., 1953; Monderer & Samet, 2002), which allocates credit to each feature by averaging contributions across all possible permutations. Let $\pi : \{1, \cdots, d\} \mapsto \{1, \cdots, d\}$ be a permutation of feature indices, $\Pi(\boldsymbol{N})$ be the set of all permutations, and $H^i(\pi)$ be the set of predecessors of feature $i$ in permutation $\pi$. The Shapley values is reformualted as follows:

$$\phi_i^{ro} = \frac{1}{d!} \sum_{\pi \in \Pi(\boldsymbol{N})} \left( v(H^i(\pi) \cup \{i\}) - v(H^i(\pi)) \right). \quad (3)$$

Then we can estimate Shapley values by Monte Carlo sampling as:

$$\phi_i^{ro} = \mathop{\mathbb{E}}_{\pi \sim p^{ro}(\pi)} \left[ v(H^i(\pi) \cup \{i\}) - v(H^i(\pi)) \right] \approx \frac{1}{M} \sum_{k=1}^{M} \left[ v(H^i(\pi) \cup \{i\}) - v(H^i(\pi)) \right], \quad (4)$$

where $p^{ro}(\pi) = 1/d!$ is the uniform distribution over all permutations.

**Least Squares Value** Shapley values can be formulated in a linear regression fashion (Ruiz et al., 1998). Specifically, Shapley values are the minimum value of a constrained weighted least squares problem and we implement the unbiased KernelSHAP (Covert & Lee, 2021) as follows:

$$\phi^{ls} = \arg\min_{\eta} \mathcal{L}(\eta) = \arg\min_{\eta} \sum_{\boldsymbol{S} \sim \omega(\boldsymbol{S})} \left( v(\boldsymbol{S}) - v(\emptyset) - \eta^T \mathbf{1}^{\boldsymbol{S}} \right)^2 \quad s.t. \ \eta^T \mathbf{1} = v(\boldsymbol{N}) - v(\emptyset) \quad (5)$$

where $\omega(\boldsymbol{S}) = \frac{d-1}{\binom{d}{|\boldsymbol{S}|}|\boldsymbol{S}|(d-|\boldsymbol{S}|)}$ is also named as Shapley kernel (Lundberg & Lee, 2017; Covert & Lee, 2021)[1]. Remarkably, the domain of $\omega(\boldsymbol{S})$ does not include the empty set $\emptyset$ and the full set $\boldsymbol{N}$.

**Proposition 1.** *The least squares value in Eq. (5) is equivalent to the semivalue in Eq. (2).*

**Remark.** *Considering that the problem in Eq. (5) is convex, we can obtain the optimal solution strictly by Karush-Kuhn-Tucker (KKT) conditions (Boyd & Vandenberghe, 2004) in Appendix A.1.*

For high dimension data, the optimization object $\mathcal{L}$ in Eq. (5) need to be approximated:

$$\mathcal{L}(\eta) = \mathop{\mathbb{E}}_{\boldsymbol{S} \sim p^{ls}(\boldsymbol{S})} \left[ v(\boldsymbol{S}) - v(\emptyset) - \eta^T \mathbf{1}^S \right]^2 \approx \frac{1}{M} \sum_{k=1}^{M} \left[ v(\boldsymbol{S}_k) - v(\emptyset) - \eta^T \mathbf{1}^{\boldsymbol{S}_k} \right]^2, \quad (6)$$

where $\boldsymbol{S}_k \sim p^{ls}(\boldsymbol{S})$ is the $k$-th sampled subset and $p^{ls}(\boldsymbol{S}) \propto \omega(\boldsymbol{S})$. This stochastic version in Eq. (6) is also named as KernelSHAP (Lundberg & Lee, 2017).

### 2.2.2 AMORTIZED ESTIMATOR

Stochastic estimators still encounter the trade-off between accuracy and efficiency. To address this, it is necessary to gain insight into the inner characteristics of models to be explained. For instance, in the case of linear models $f(\boldsymbol{x}) = \omega^T \boldsymbol{x} + b$ with value function $v(\boldsymbol{S}) = \omega^T (\mathbf{1}^{\boldsymbol{S}} \cdot \boldsymbol{x}) + b$, the Shapley values can be easily computed in closed form as $\phi_i = \omega_i \boldsymbol{x}_i$ with linear complexity $\mathcal{O}(d)$. Previous

---

[1]See Appendix A.8 for unbiased KernelSHAP estimation.

Table 1: Unified Stochastic Estimator.

| Estimator | Prob. $p^i(\boldsymbol{S})$ | Coef. $a_{\boldsymbol{S}}^i$ | Trans. $\boldsymbol{T}$ | Bias $\boldsymbol{b}$ |
|---|---|---|---|---|
| Random Order Value & Semivalue | $\left[\binom{d}{\|\boldsymbol{S}\|}(\|\boldsymbol{S}\|\mathbb{I}_{i\in\boldsymbol{S}} + (d-\|\boldsymbol{S}\|)\mathbb{I}_{i\notin\boldsymbol{S}})\right]^{-1}$ | $2\mathbb{I}_{i\in\boldsymbol{S}} - 1$ | $\boldsymbol{I}$ | $\boldsymbol{0}$ |
| Least Squares Value | $\left[\gamma\binom{d}{\|\boldsymbol{S}\|}(d-\|\boldsymbol{S}\|)\|\boldsymbol{S}\|\right]^{-1}$ | $\mathbb{I}_{i\in\boldsymbol{S}}$ | $\gamma(d\boldsymbol{I} - \boldsymbol{J})$ | $\frac{v(\boldsymbol{N})-v(\emptyset)}{d}\boldsymbol{1}$ |
| **Sim-Semivalue** (Ours) | $\left[\gamma\binom{d}{\|\boldsymbol{S}\|}(d-\|\boldsymbol{S}\|)\|\boldsymbol{S}\|\right]^{-1}$ | $\gamma(d-\|\boldsymbol{S}\|)\mathbb{I}_{i\in\boldsymbol{S}}$ $-\gamma\|\boldsymbol{S}\|\mathbb{I}_{i\notin\boldsymbol{S}}$ | $\boldsymbol{I}$ | $\frac{v(\boldsymbol{N})-v(\emptyset)}{d}\boldsymbol{1}$ |

works have also explored model-specific Shapley values estimators for various model structures, such as tree-based models (Lundberg et al., 2020) and neural networks (Shrikumar et al., 2017; Chen et al., 2019b; Wang et al., 2021; Chen et al., 2023b). However, these methods require subtle design and may even necessitate special modules or training techniques. To tackle this, FastSHAP (Jethani et al., 2021) trains an amortized parametric function $g(\boldsymbol{x};\theta) : \mathcal{X} \mapsto \mathbb{R}^d$ to estimate Shapley values by penalizing predictions according to the weighted least squares loss in Eq. (5):

$$\theta = \arg\min_{\theta} \mathbb{E}_{\boldsymbol{x}\in\mathcal{X}} \mathbb{E}_{\boldsymbol{S}\sim p^{ls}(\boldsymbol{S})} \left[v(\boldsymbol{S}) - v(\emptyset) - g(\boldsymbol{S};\theta)^T\boldsymbol{1}^{\boldsymbol{S}}\right]^2 \quad s.t. \ g(\boldsymbol{x};\theta)^T\boldsymbol{1} = v(\boldsymbol{N}) - v(\emptyset). \quad (7)$$

To optimize the constrained problem in Eq. (7), Jethani et al. (2021) adjusts predictions using additive efficient normalization (Ruiz et al., 1998) or optimizes with a penalty on the efficiency gap.

### 2.3 UNIFIED PERSPECTIVE

Numerous researches have been conducted to accelerate the stochastic estimation of Shapley values. However, interconnections among these methods are not elucidated clearly. In this section, we put forth a unified perspective for Shapley value estimation. This perspective not only unveils the exact disparities among these methods but also presents a fresh direction for future research endeavors.

**Definition 2.** *Unified Stochastic Estimator is defined as the linear transformation of the values obtained from sampled subsets $\boldsymbol{S}$ as:*

$$\phi^{uni} = \boldsymbol{T}\tilde{\phi} + \boldsymbol{b}, \quad \tilde{\phi}_i = \mathbb{E}_{\boldsymbol{S}\sim p^i(\boldsymbol{S})}\left[a_{\boldsymbol{S}}^i v(\boldsymbol{S})\right] \approx \underbrace{\frac{1}{M}\sum_{k=1}^M a_{\boldsymbol{S}}^i v(\boldsymbol{S}_k)}_{\text{Importance sampling}}, \quad (8)$$

*where $\boldsymbol{T} \in \mathbb{R}^{d\times d}$ and $\boldsymbol{b} \in \mathbb{R}^d$ are transformation parameters, and $a_{\boldsymbol{S}}^i$ is the coefficient of subset $\boldsymbol{S}$.*

**Remark.** *Three ways of estimating Shapley values are special cases of Definition 2 as illustrated in Table 1. The detailed analysis is provided in the following.*

**Semivalue & Random Order Value** It is apparent that the semivalue in Eq. (2) and the random order value in Eq. (4) are equivalent. This arises from the fact that the number of permutations equals $(d - \boldsymbol{S} - 1)!\boldsymbol{S}!$ given subset $\boldsymbol{S}$ and feature $i$ being positioned after features in $\boldsymbol{S}$. The semivalue can be reformulated under the framework of Definition 2 as:

$$\phi_i^{sv} = \boldsymbol{T}_i^{sv}\tilde{\phi}^{sv} + \boldsymbol{b}_i^{sv} = \sum_{\boldsymbol{S}\subset\boldsymbol{N}} p^i(\boldsymbol{S})a_{\boldsymbol{S}}^{sv}v(\boldsymbol{S}) = \underbrace{\sum_{\boldsymbol{S}\subset\boldsymbol{N}\setminus\{i\}} \frac{v(\boldsymbol{S}\cup\{i\}) - v(\boldsymbol{S})}{\binom{d}{\|\boldsymbol{S}\|}(d-\|\boldsymbol{S}\|)}}_{\text{Shapley values}}. \quad (9)$$

**Least Squares Value** This value is the closed form solution by Lagrange multiplier method, which is also unified form under Definition 2 as:

$$\phi^{ls} = \boldsymbol{T}^{ls}\tilde{\phi}^{ls} + \boldsymbol{b}^{ls} = (d\boldsymbol{I} - \boldsymbol{J})\tilde{\phi}^{ls} + \frac{v(\boldsymbol{N}) - v(\emptyset)}{d}\boldsymbol{1} \quad (10)$$

$$\tilde{\phi}_i^{ls} = \sum_{\boldsymbol{S}\subsetneq\boldsymbol{N}\setminus\emptyset} p^i(\boldsymbol{S})a_{\boldsymbol{S}}^{ls}v(\boldsymbol{S}) = \sum_{\boldsymbol{S}\subsetneq\boldsymbol{N}\setminus\emptyset} \frac{v(\boldsymbol{S})\mathbb{I}_{i\in\boldsymbol{S}}}{\binom{d}{\|\boldsymbol{S}\|}(d-\|\boldsymbol{S}\|)\|\boldsymbol{S}\|}, \quad (11)$$

where $\boldsymbol{J}$ is the matrix with all ones. See detailed derivation in Appendix A.1. Eq. (10) reveals that the essence of least squares value is, in fact, another instance of direct sampling rather than the minimization of least squares loss.

Model-agnostic stochastic estimators still suffer from the trade-off between accuracy and efficiency because of the unavoidable exponential complexity of potential feature subsets. Recent amortized estimators (Jethani et al., 2021) further accelerate the computation by leveraging the inner structure of the model to be explained, requiring only a single forward pass to estimate Shapley values. Anologously to Definition 2, we derive the unified form of amortized estimators as follows:

**Definition 3.** *Unified Amortized Estimator is defined as a learnable parametric function* $g(\boldsymbol{x}; \theta)$ : $\mathcal{X} \mapsto \mathbb{R}^d$ *to fit towards true (or estimated) Shapley values* $\phi_{\boldsymbol{x}}$:

$$\theta = \arg \min_{\theta} \mathop{\mathbb{E}}_{\boldsymbol{x} \in \mathcal{X}} \left[ \|g(\boldsymbol{x}; \theta) - \phi_{\boldsymbol{x}}\|_M^2 \right], \tag{12}$$

*where* $\boldsymbol{M} \in \mathbb{S}^+$ *is the metric matrix.*

To simplify the notation, given the order of non-empty proper subset $(\boldsymbol{S}_1, \cdots, \boldsymbol{S}_n)$ and $n = 2^d - 2$, we define the value vector $\boldsymbol{v} = (v(\boldsymbol{S}_1), \cdots, v(\boldsymbol{S}_n)) \in \mathbb{R}^n$, the indicator matrix $\boldsymbol{X} = (\mathbf{1}^{\boldsymbol{S}_1}, \cdots, \mathbf{1}^{\boldsymbol{S}_n})^T \in \{0, 1\}^{n \times d}$, and the weight matrix $\boldsymbol{W} = diag(\omega(\boldsymbol{S}_1), \cdots, \omega(\boldsymbol{S}_n))$, where $\omega$ is the Shapley kernel. If we let $\phi_{\boldsymbol{x}}$ equal to or approximate to the value of $(\boldsymbol{X}^T \boldsymbol{W} \boldsymbol{X})^{-1} \boldsymbol{X}^T \boldsymbol{W} \boldsymbol{v}$ and $\boldsymbol{M} = \boldsymbol{X}^T \boldsymbol{W} \boldsymbol{X}$, Eq. (12) degenerates to the least squares loss of FastSHAP (Jethani et al., 2021) as[2]:

$$\mathcal{L} = \mathop{\mathbb{E}}_{\boldsymbol{x} \in \mathcal{X}} \left[ \left\|g(\boldsymbol{x}; \theta) - (\boldsymbol{X}^T \boldsymbol{W} \boldsymbol{X})^{-1} \boldsymbol{X}^T \boldsymbol{W} \boldsymbol{v} \right\|_{\boldsymbol{X}^T \boldsymbol{W} \boldsymbol{X}}^2 \right] = \mathop{\mathbb{E}}_{\boldsymbol{x} \in \mathcal{X}} \left[ \|\boldsymbol{X} g(\boldsymbol{x}; \theta) - \boldsymbol{v}\|_{\boldsymbol{W}}^2 \right] + C. \tag{13}$$

Because the fitting target of FastSHAP is biased, the predictions of FastSHAP need to be rectified by additive efficient normalization (Ruiz et al., 1998) as: $g(\boldsymbol{x}; \theta) \leftarrow \frac{d\boldsymbol{I} - \boldsymbol{J}}{d} g(\boldsymbol{x}; \theta) + \frac{v(\boldsymbol{N}) - v(\emptyset)}{d} \mathbf{1}$.

## 2.4 SIMSHAP

Based on discussion in Section 2.3, we propose a straightforward amortized estimator named as **SimSHAP**. Different from the special metric matrix in Fast-SHAP, we simply select the identity matrix $\boldsymbol{M} = \boldsymbol{I}$ as the metric matrix. By combining advantages of semivalue and least squares value, we propose a new sampling method dubbed as **Sim-Semivalue** shown in the last row of Table 1. We sample subsets $\boldsymbol{S}$ according to the least squares value distribution $p^{ls}(\boldsymbol{S})$, which is irrelevant to index $i$ and is more friendly to parallel computation. Besides, we assign the coefficient as $a_{\boldsymbol{S}}^i = \gamma(d - |\boldsymbol{S}|)\mathbb{I}_{i \in \boldsymbol{S}} - \gamma|\boldsymbol{S}|\mathbb{I}_{i \notin \boldsymbol{S}}$. The final algorithm of SimSHAP is formulated in Algorithm 1.

---

**Algorithm 1:** SimSHAP training.

**Input:** Value function $v$, learning rate $\alpha$, number of subsets $M$.
**Output:** SimSHAP explainer $g(\boldsymbol{x}; \theta)$.

1  Initialize random weights $\theta$;
2  **while** *not converged* **do**
3      Sample the input $\boldsymbol{x} \in \mathcal{X}$;
4      Sample $M$ subsets $\{\boldsymbol{S}_i | \boldsymbol{S}_i \sim p^{ls}(\boldsymbol{S})\}$;
5      Compute the estimated Shapley values as:
    $\hat{\phi} = \frac{1}{M} \sum_{k=1}^{M} \gamma((d - |\boldsymbol{S}|)\mathbb{I}_{i \in \boldsymbol{S}} - |\boldsymbol{S}|\mathbb{I}_{i \notin \boldsymbol{S}}) v(\boldsymbol{S_k}) + \frac{v(\boldsymbol{N}) - v(\emptyset)}{d} \mathbf{1}$;
6      Compute $\mathcal{L} = \|g(x; \theta) - \hat{\phi}\|_2^2$;
7      Update the parameters as: $\theta \leftarrow \theta - \alpha \nabla_{\theta} \mathcal{L}$;

---

It is obvious that the expectation of the fitting target of SimSHAP is unbiased as:

$$\mathbb{E}[\phi_{\boldsymbol{x}}] = \boldsymbol{T} \sum_{\boldsymbol{S} \subsetneq \boldsymbol{N} \setminus \emptyset} p^{ls}(\boldsymbol{S}) a_{\boldsymbol{S}} v(\boldsymbol{S}) + \boldsymbol{b} = \sum_{\boldsymbol{S} \subsetneq \boldsymbol{N} \setminus \emptyset} \frac{[(d - |\boldsymbol{S}|)\mathbb{I}_{i \in \boldsymbol{S}} - |\boldsymbol{S}|\mathbb{I}_{i \notin \boldsymbol{S}}] v(\boldsymbol{S})}{\binom{d}{|\boldsymbol{S}|}(d - |\boldsymbol{S}|)|\boldsymbol{S}|} + \frac{v(\boldsymbol{N}) - v(\emptyset)}{d} \mathbf{1}$$

$$= \sum_{\boldsymbol{S} \subset \boldsymbol{N} \setminus \{i\}} \frac{v(\boldsymbol{S} \cup \{i\}) - v(\boldsymbol{S})}{\binom{d}{|\boldsymbol{S}|}|\boldsymbol{S}|(d - |\boldsymbol{S}|)}. \tag{14}$$

If the capacity of $g(\boldsymbol{x}; \theta)$ is large enough, SimSHAP will fit towards the true Shapley values as accurate as possible. Main differences between FastSHAP and SimSHAP are compared in Table 2. The time complexities of these two algorithms are comparable, and a detailed analysis can be found in Appendix A.10.

---

[2]For simplicity, we only derived the case where $\phi_{\boldsymbol{x}}$ is true value.

Table 2: Unified Amortized Estimator.

| Estimator | $\mathbb{E}[\phi_{\boldsymbol{x}}]$ | Metric Matrix $M$ | Normalization |
|---|---|---|---|
| FastSHAP | $(\boldsymbol{X}^T \boldsymbol{W} \boldsymbol{X})^{-1} \boldsymbol{X}^T \boldsymbol{W} \boldsymbol{v}$ | $\boldsymbol{X}^T \boldsymbol{W} \boldsymbol{X}$ | $\frac{d\boldsymbol{I} - \boldsymbol{J}}{d}(\cdot) + \frac{v(\boldsymbol{N}) - v(\emptyset)}{d}\mathbf{1}$ |
| **SimSHAP** (Ours) | $\frac{d\boldsymbol{I} - \boldsymbol{J}}{d-1}\boldsymbol{X}^T \boldsymbol{W} \boldsymbol{v} + \frac{v(\boldsymbol{N}) - v(\emptyset)}{d}\mathbf{1}$ | $\boldsymbol{I}$ | (unbiased) |

## 3 RELATED WORK

**Explanation by Shapley Values.**   Numerous attemps (Zhang & Zhu, 2018; Gilpin et al., 2018; Zhang et al., 2021; Bodria et al., 2021) have explored how to reveal the complex mechanisms of black-box models. To ensure the fidelity, the explanation methods should establish a stable mapping between abstract feature representations and concrete human concepts. Shapley values are first proposed in cooperative game theory (Shapley et al., 1953) to quantify the contribution of each player in a coalition. Theoretically, the Shapley values are the unique values that satisfy the four fairness axioms: efficiency, symmetry, linearlity, and dummy player (Shapley et al., 1953; Dubey et al., 1981; Lipovetsky & Conklin, 2001). Recently, Shapley values have been introduced into machine learning to explain the predictions of black-box models, such as DNNs. However, machine learning models are not originally designed for cooperative games and the value functions remain to be well defined (Covert et al., 2021; Frye et al., 2021; Aas et al., 2021). Proper choices of value functions are crucial to avoid information leakage and align explanations with human concepts (Rong et al., 2022; Jethani et al., 2023). Besides, exact Shapley values are usually intractable to compute due to the exponential complexity of the data dimension, which gives rise to the development of efficient estimator strategies, which can be categorized into **model-agnostic** and **model-specific** methods.

**Model-agnostic Estimation.**   Shapley values are linear transformation of values with respect to all feature combinations. To relieve the exponential complexity of exactly computing Shapley values, various works have been proposed to estimate Shapley values by sampling subsets of potential combinations. **Semivalues** (Dubey et al., 1981) are the classic form of Shapley values and ApproSemivalue (Castro et al., 2009) is a polynomial-time approximation of semivalue by importance sampling. **Random order values** (Monderer & Samet, 2002) reformulate Shapley values as the expectation of marginal contributions with random permutations, which can be efficiently estimated by uniformly sampling feature permutations (Castro et al., 2009; Strumbelj & Kononenko, 2010). Besides, **least sqares values** (Charnes et al., 1988; Ruiz et al., 1998; Lipovetsky & Conklin, 2001) discover that Shapley values are the solution to a least squares problem. KernelSHAP (Lundberg & Lee, 2017; Covert & Lee, 2021) utilizes this characteristic to estimate Shapley values by solving a linear regression problem with sampled subsets. Since numerous works have been proposed to accelerate the estimation of Shapley values, we wonder how large the differences are between different estimation strategies. Therefore, we investigate the unified stochastic estimator in this work.

**Model-specific Estimation.**   To further improve the efficiency of Shapley value estimation, some works have been proposed to slim the redundant subsets by exploiting the specific structures of machine learning models. For linear models, the complexity of computing Shapley values can be reduced to linear time by the linearity of models (Lundberg & Lee, 2017). For tree-based models, TreeExplainer (Lundberg et al., 2020) computes local explanations based on exact Shapley values in polynomial time. For deep neural networks, DeepLIFT (Shrikumar et al., 2017) efficiently compute the attributions by backpropagating the contributions of all neurons to every feature of the input. ShapleyNet (Wang et al., 2021) and HarsanyiNet (Chen et al., 2023b) propose special network structures to compute Shapley values in a single forward pass. There are also methods (Chen et al., 2019a; Teneggi et al., 2022) that achieve exact, finite-sample approximation of Shapley values by making distribution assumptions. However, it is untrival to extend these methods to other network structures and distribution. To further improve the flexibility, recent works (Schwarzenberg et al., 2021; Jethani et al., 2021; Chuang et al., 2023) employ a learnable parametric function to approximate explanations.[3] Surprisingly, we find that amortized methods can also be formulated into a unified framework, based on which we further propose SimSHAP.

## 4 EXPERIMENTS

### 4.1 STRUCTURED DATA EXPERIMENTS

We assess the performance of SimSHAP by comparing it to popular baselines. Our evaluation begins with an examination of its accuracy and reliability across various tabular datasets, involving

---

[3]See Appendix A.9 for detailed comparison with  Schwarzenberg et al. (2021); Chuang et al. (2023)

Figure 2: SimSHAP estimation accuracy across tabular datasets.

a comparison of model outputs to ground truth Shapley values. Later on, we employ a well-known image dataset to evaluate image explanation through classical metrics.

### 4.1.1 EXPERIMENTAL DETAILS

**Dataset Description** We test SimSHAP's accuracy through different tabular datasets. We use `census`, `news`, and `bankruptcy` for in-depth evaluation. The `census` dataset was sourced from 1994 United States Census database. It contains 12 features, with the label indicating whether an individual's income exceeds $50,000/year (Kohavi et al., 1996). The `news` dataset contains 60 numerical features related to articles published on Mashable over a two-year period, with the label indicating whether the share count exceeds the median value of 1400 (Fernandes et al., 2015). The `bankruptcy` dataset was obtained from the Taiwan Economic Journal for the years 1999 to 2009. It contains 96 features and the label indicates whether the company went bankrupt (Liang et al., 2016). These datasets are each split 80/10/10 for training, validation and testing.

**Implementation Details** For original models $f : \mathcal{X} \mapsto \mathcal{Y}$, we opt for tree-based methods across all datasets. Following the approach of FastSHAP (Jethani et al., 2021), we train neural networks as surrogate models and employ these models as the value function for explanation model training. The SimSHAP explainer model $\phi_{fast}(\boldsymbol{x}, \boldsymbol{y}; \theta)$ is implemented as an MLP $g(\boldsymbol{x}, \theta) : \mathcal{X} \mapsto \mathbb{R}^d \times \mathcal{Y}$ that outputs a vector of Shapley values for each $y \in \mathcal{Y}$, following the methodology as FastSHAP (Jethani et al., 2021). In order to balance training speed and accuracy, we choose 64 samples for each data $\boldsymbol{x}$, use pair sampling, and train for 1000 epochs for each tabular dataset. Further details concerning our code and hyperparameter selection can be found in our codebase and Appendix A.5. For a comprehensive evaluation, we compared SimSHAP against several baselines, the majority of which are non-amortized iterative methods. Specifically, We compare to KernelSHAP (Lundberg & Lee, 2017) and its improvement utilizing pair sampling (Covert & Lee, 2021). Additionally, we compare to permutation sampling, as well as an improvement method using antithetical sampling (Mitchell et al., 2022). Last, we compare to FastSHAP. To compute the distance, we run KernelSHAP to convergence for a given threshold to compute the ground truth Shapley values.

### 4.1.2 QUANTITATIVE EXPERIMENTS

We choose 3 datasets for quantitative assessments. As we are able to get ground truth Shapley values for tabular datasets, we can calculate the accuracy among different methods. $l_1$ and $l_2$ distances are used between estimations and ground truth for comparison. As illustrated in Fig. 2, we compare SimSHAP with non-amortized methods and FastSHAP. All explanation methods are estimated with the same surrogate model. Results demonstrate that SimSHAP achieves comparable accuracy to FastSHAP across different datasets. In the case of non-amortized methods, pair sampling proves to be more effective in two KernelSHAP estimations, and we adopted the technique in SimSHAP. We can also address this issue in formula analysis, as the weights of both **Semivalue** and **Least Square Value** are symmetric and is higher for smaller values and larger values. In addition, antithetical sampling is also effective than Monte Carlo permutation method. We also conduct experiments on exploring the effect of different hyperparameters on the accuracy and convergence rate of SimSHAP. Please refer to Appendix A.6 for more details.

## 4.2 IMAGE DATA EXPERIMENTS

The intricate nature and high dimensionality of image data pose significant challenges for explanation methods. In this section, we assess the performance of SimSHAP on CIFAR-10 (Krizhevsky et al., 2009) dataset and compare it with widely-used baseline methods.

### 4.2.1 EXPERIMENTAL DETAILS

**Dataset Description** We evaluate SimSHAP's explainability on high-dimensional data using **CIFAR-10** dataset. It contains 60000 $32 \times 32$ images belonging to 10 classes. For experiments,

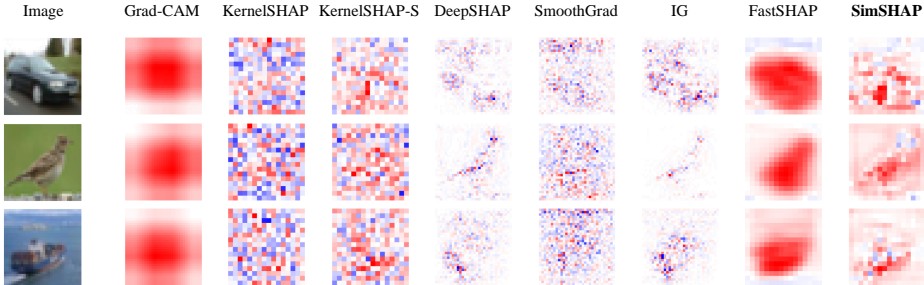

Figure 3: Comparison of different methods on randomly-chosen images in CIFAR-10.

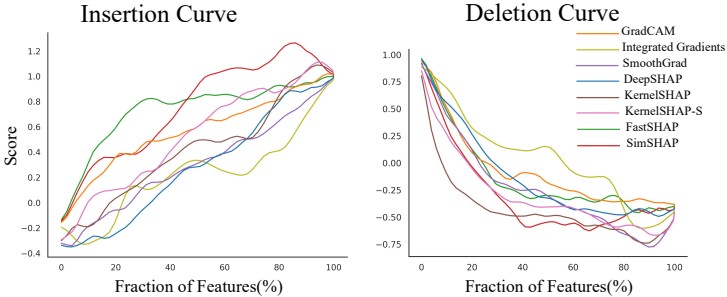

Figure 4: Mean Insertion and Deletion score curves for different methods on CIFAR-10 dataset

we used 50000 images for training and equally split the rest 10000 images for validation and testing.

**Implementation Details** For original model and surrogate models, we adopt the ResNet-18 structure (He et al., 2016) for modeling. Similar to FastSHAP (Jethani et al., 2021), we assign image attributions to $16 \times 16$ regions, with each element corresponding to a $2 \times 2$ superpixel. We employ a U-net (Ronneberger et al., 2015) structure as the SimSHAP explainer model. For baseline methods, we compare our SimSHAP method with various Shapley value estimators. KernelSHAP and DeepSHAP (Lundberg & Lee, 2017) follow the principle of Shapley Additive Explanation, providing a unified measure of feature importance. KernelSHAP is a model-agnostic method, while DeepSHAP is designed for neural networks which leverages the nature of neural networks for better performance. In addition, we use Kernelshap-S method as the same as FastSHAP (Jethani et al., 2021) which employs a surrogate model as the value function. We also compare SimSHAP with other gradient-based methods, including Integrated Gradients (Sundararajan et al., 2017), Smooth-Grad (Smilkov et al., 2017), and GradCAM (Selvaraju et al., 2017). In our SimSHAP implementation, we choose 8 samples per image, use pair sampling, and conduct training for 500 epochs.

### 4.2.2 QUALITATIVE EXPERIMENTS

We commence with the presentation of three images selected randomly from the test set and their respective attributions generated by different methods, shown in Fig. 3. SimSHAP can find important regions, like GradCAM and FastSHAP, but it can better protray the contours of the main object. It also has the advantage of DeepSHAP or Integrated Gradients, applying different rate of importance in the object. For example, the SimSHAP explanation on the second row shows that the model focuses on the body and the tail of the bird rather than the head to classify this image as "Bird". In comparison, KernelSHAP and KernelSHAP-S give more chaos attributions, and SmoothGrad is not able to find the main object. With a qualitative evaluation one may conclude that SimSHAP is a promising method for image explanation. Next, we show quantitative experiments to further validate this conclusion.

### 4.2.3 QUANTITATIVE EXPERIMENTS

It's hard to evaluate the accuracy of image explanation because of the computational complexity of high-dimensional data. Instead, we use **Insertion / Deletion metrics** (Petsiuk et al., 2018). These scores measure how well explanations identify informative regions within the image. The *Deletion* metric aims to find how the model's decision changes as more and more important pixels are removed, while the *Insertion* metric takes the opposite way. Scores are collected as the target class output of the masked image. At the end of the masking process, scores are normalized so that it starts with 0/1 and ends with 1/0 for Insertion/Deletion metrics. We delete/insert pixels sequentially in the descending order of attributions, plot the output curve, and calculate the area under the curve

(AUC). As recommended by Petsiuk et al. (2018), we set the baseline image as zero for deletions, and Gaussian-blurred image for insertions.

Fig. 4 shows the mean insertion score and deletion score curves. For the insertion curves, SimSHAP outperforms Fast-SHAP as it attains the highest values at the end of the curve. For the deletion curves, SimSHAP achieves similar results as KernelSHAP-S, with a relatively rapid decrease and consistent low scores.

Following the protocol (Jethani et al., 2021), a comprehensive comparison is provided in Table 3. SimSHAP exhibits the best insertion AUC and the second best deletion AUC. For Insertion AUC, Fast-SHAP performs comparably to SimSHAP in terms of insertion AUC, while Grad-CAM and KernelSHAP-S also exhibit effectiveness. For Deletion AUC, KernelSHAP outperforms all other methods, albeit with a high standard deviation. Given that a reliable Shapley value estimator should perform well on both metrics, SimSHAP emerges as a compelling choice.

Table 3: Insertion / Deletion metrics on CIFAR-10.

| | CIFAR-10 | |
|---|---|---|
| | Insertion AUC↑ | Deletion AUC↓ |
| FastSHAP | **0.748** (±0.082) | -0.133 (±0.055) |
| GradCAM | 0.563 (±0.044) | -0.075 (±0.034) |
| IG | 0.241 (±0.051) | 0.033 (±0.052) |
| SmoothGrad | 0.318 (±0.052) | -0.246 (±0.103) |
| DeepSHAP | 0.291 (±0.101) | -0.140 (±0.173) |
| KernelSHAP | 0.430 (±0.064) | **-0.443** (±0.157) |
| KernelSHAP-S | 0.542 (±0.052) | **-0.305** (±0.152) |
| **SimSHAP (Ours)** | **0.757** (±0.117) | -0.302 (±0.063) |

Table 4: Inference time(s) / training time(min) on both tabular and image datasets.

| | | Census | News | Bank | CIFAR-10 |
|---|---|---|---|---|---|
| **Inference** | FastSHAP | 0.004 | 0.004 | 0.005 | 0.090 |
| | GradCAM | - | - | - | 1.907 |
| | IG | - | - | - | 25.486 |
| | KernelSHAP | 4.438 | 66.410 | 96.615 | 1440.595 |
| | KernelSHAP-S | 43.560 | 67.002 | 101.053 | 1418.292 |
| | **SimSHAP (Ours)** | **0.002** | **0.001** | **0.001** | **0.086** |
| **Train** | FastSHAP | 11.098 | 16.223 | 2.005 | **97.548** |
| | **SimSHAP (Ours)** | **7.466** | **7.603** | **1.206** | 324.100 |

### 4.2.4 SPEED EVALUATION

We present results of training speed and inference speed for different methods on both tabular and image datasets (see Table 4). We can only evaluate SimSHAP and FastSHAP for training speed. Results reveal that SimSHAP requires less training time (but more epochs) than FastSHAP in order to reach FastSHAP's accuracy on tabular datasets, but requires more time on image datasets. That's mostly because of the requirement of number of mask is larger for SimSHAP, and the data For inference speed in tabular datasets, gradient-based methods are not applicable as we employ tree-based models as the original model. SimSHAP outperforms all the baseline methods in terms of inference speed. It is slightly faster than FastSHAP because there is no need for SimSHAP to compute normalize the output of the explainer model for the effeciency constraint. Gradient-based methods are slightly slower, while KernelSHAP and KernelSHAP-S require more time due to backward computation and a large number of iterations during inference. Lastly, we test the robustness of SimSHAP with limited data. These experiments reveal that acceptable performance can be achieved with just 20% of the data, with scores becoming more stable as more data is involved in training. We also conduct experiments on exploring the effect of different hyperparameters on the accuracy and convergence rate of SimSHAP, including the number of samples, the number of epochs, and the choice of learning rate. See Appendix A.6 for more details.

## 5 DISCUSSION

The issue of interpretability in black box models is crucial for enhancing model performance and gaining user trust. Shapley values provide a reliable and interpretable attribution method based on axiomatic principles. However, the computational complexity of Shapley values hinders their practical applicability. This study aims to clarify the inherent connections between current random estimators and the latest amortized estimators, and proposes a unified perspective on estimation. Building upon this, we introduce SimSHAP, a simple and efficient estimator. However, we acknowledge that there are unresolved issues in this study, with the most significant concern being the design problem of amortized models, as accurate fitting relies on appropriate model design. While there have been equivalence studies conducted on general MLP networks (Villani & Schoots, 2023), these findings are insufficient when applied to complex CNNs and ViTs. We anticipate that future research will comprehensively investigate this aspect.

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

# A   APPENDIX

## A.1   PROOF OF EQUIVALENCE BETWEEN SEMIVALUE AND LEAST SQUARES VALUE

This proposition was first carefully studied by Charnes et al. (1988), and here we review it.

*Proof.* Before starting the proof, we first investigate the characteristics of coefficients $\omega(\boldsymbol{S}) = \frac{d-1}{\binom{d}{|\boldsymbol{S}|}|\boldsymbol{S}|(d-|\boldsymbol{S}|)}$ and find the equations as follows:

$$\sum_{\boldsymbol{S} \subsetneq \boldsymbol{N} \backslash \emptyset, i \in \boldsymbol{S}} \omega(\boldsymbol{S}) \phi^T \mathbf{1}^{\boldsymbol{S}} = A\phi_i + \sum_{j \neq i} \left( A - \frac{d-1}{d}\phi_j \right) \tag{15}$$

$$\text{where } A = \frac{d-1}{d}\left( \frac{1}{d-1} + \cdots + 1 \right). \tag{16}$$

Considering that the problem in Eq. (5) is convex, we can obtain the optimal solution by Karush-Kuhn-Tucker (KKT) conditions (Boyd & Vandenberghe, 2004). We first formnulate the Lagrangian function as:

$$L(\lambda, \phi) = \sum_{\boldsymbol{S} \subsetneq \boldsymbol{N} \backslash \emptyset} \omega(\boldsymbol{S}) \left( v(\boldsymbol{S}) - v(\emptyset) - \phi^T \mathbf{1}^{\boldsymbol{S}} \right)^2 + \lambda (v(\boldsymbol{N}) - v(\emptyset) - \phi^T \mathbf{1}). \tag{17}$$

Then we can obtain the optimal solution by KKT conditions as:

$$\frac{\partial L}{\partial \phi} = 2 \sum_{\boldsymbol{S} \subsetneq \boldsymbol{N}, i \in \boldsymbol{S}} \omega(\boldsymbol{S}) \left( \phi^T \mathbf{1}^{\boldsymbol{S}} - v(\boldsymbol{S}) + v(\emptyset) \right) - \lambda = 0, \;\; i = 1, 2, \cdots, d \tag{18}$$

$$\frac{\partial L}{\partial \lambda} = v(\boldsymbol{N}) - v(\emptyset) - \phi^T \mathbf{1} = 0. \tag{19}$$

Sum the both sides of $d$ equations in Eq. (18) and we can obtain $\lambda$:

$$\lambda = \frac{2}{d} \sum_{i=1}^{d} \sum_{\boldsymbol{S} \subsetneq \boldsymbol{N}, i \in \boldsymbol{S}} \omega(\boldsymbol{S}) \left( \phi^T \mathbf{1}^{\boldsymbol{S}} - v(\boldsymbol{S}) + v(\emptyset) \right)$$

$$= \frac{2}{d} \left[ \sum_{i=1}^{d} \sum_{\boldsymbol{S} \subsetneq \boldsymbol{N}, i \in \boldsymbol{S}} \omega(\boldsymbol{S}) \left( \phi^T \mathbf{1}^{\boldsymbol{S}} \right) \right] - \frac{2}{d} \left[ \sum_{i=1}^{d} \sum_{\boldsymbol{S} \subsetneq \boldsymbol{N}, i \in \boldsymbol{S}} \omega(\boldsymbol{S}) \left( v(\boldsymbol{S}) - v(\emptyset) \right) \right]$$

$$= \frac{2}{d} \left[ \sum_{i=1}^{d} \left( \frac{d-1}{d}\phi_i + B\phi^T \mathbf{1} \right) \right] - \frac{2}{d} \left[ \sum_{i=1}^{d} \sum_{\boldsymbol{S} \subsetneq \boldsymbol{N}, i \in \boldsymbol{S}} \omega(\boldsymbol{S}) \left( v(\boldsymbol{S}) - v(\emptyset) \right) \right]$$

$$= \frac{2}{d} \left[ \left( dB - \frac{d-1}{d} \right) (v(\boldsymbol{S}) - v(\emptyset)) \right] - \frac{2}{d} \left[ \sum_{i=1}^{d} \sum_{\boldsymbol{S} \subsetneq \boldsymbol{N}, i \in \boldsymbol{S}} \omega(\boldsymbol{S}) \left( v(\boldsymbol{S}) - v(\emptyset) \right) \right], \tag{20}$$

where $B = A - \frac{d-1}{d}$ in Eq. (15). Finally we substitute $\lambda$ into Eq. (18) and obtain the optimal solution $\phi^*$ as:

$$\frac{\lambda}{2} = \sum_{\boldsymbol{S} \subsetneq \boldsymbol{N}, i \in \boldsymbol{S}} \omega(\boldsymbol{S}) \left( \phi^T \mathbf{1}^{\boldsymbol{S}} - v(\boldsymbol{S}) + v(\emptyset) \right)$$

$$\Rightarrow \phi_i = d \left[ \sum_{\boldsymbol{S} \subsetneq \boldsymbol{N}, i \in \boldsymbol{S}} \frac{(d - |\boldsymbol{S}| - 1)!(|\boldsymbol{S}| - 1)!}{d!} v(\boldsymbol{S}) \right]$$

$$- \left[ \sum_{j=1}^{d} \sum_{\boldsymbol{S} \subsetneq \boldsymbol{N}, j \in \boldsymbol{S}} \frac{(d - |\boldsymbol{S}| - 1)!(|\boldsymbol{S}| - 1)!}{d!} v(\boldsymbol{S}) \right] + \frac{1}{d}(v(\boldsymbol{N}) - v(\emptyset))$$

$$= \sum_{\boldsymbol{S} \subsetneq \boldsymbol{N}, i \in \boldsymbol{S}} \frac{(d - |\boldsymbol{S}|)!(|\boldsymbol{S}| - 1)!}{d!} v(\boldsymbol{S}) - \sum_{\boldsymbol{S} \subsetneq \boldsymbol{N} \backslash \emptyset, i \notin \boldsymbol{S}} \frac{(d - |\boldsymbol{S}| - 1)!(|\boldsymbol{S}|)!}{d!} v(\boldsymbol{S})$$

$$+ \frac{1}{d}(v(\boldsymbol{N}) - v(\emptyset)) \tag{21}$$

Above all, we can proof that the solution of Eq. (5) is equivalent to the Shapley values. $\qquad\square$

For the sake of simplicity, we rewrite Eq. (21) in the matrix form below. We follow the matrix definitions used in the main text here. given the order of non-empty proper subset $(\boldsymbol{S}_1, \cdots, \boldsymbol{S}_n)$ and $n = 2^d - 2$, we define the value vector $\boldsymbol{v} = (v(\boldsymbol{S}_1), \cdots, v(\boldsymbol{S}_n)) \in \mathbb{R}^n$, the indicator matrix $\boldsymbol{X} = (\mathbf{1}^{\boldsymbol{S}_1}, \cdots, \mathbf{1}^{\boldsymbol{S}_n})^T \in \{0,1\}^{n \times d}$, and the weight matrix $\boldsymbol{W} = diag(\omega(\boldsymbol{S}_1), \cdots, \omega(\boldsymbol{S}_n))$, where $\omega$ is the Shapley kernel. Firstly, we find that the matrix $\boldsymbol{X}, \boldsymbol{W}$ has some very nice properties as:

$$\boldsymbol{X}^T \boldsymbol{W} \boldsymbol{X} = \frac{d-1}{d} \boldsymbol{I} + B \boldsymbol{J}$$

$$(\boldsymbol{X}^T \boldsymbol{W} \boldsymbol{X})^{-1} = \frac{d}{d-1} \boldsymbol{I} - C \boldsymbol{J}$$

where $B = \frac{d-1}{d} \left( \frac{1}{d-1} + \cdots + 1 \right)$, $C = \frac{d^2 B}{(d-1)(d^2 B + d - 1)}$. Secondly, we rewrite the Lagrangian function Eq. (17) in the matrix form as:

$$L(\lambda, \phi) = \|\boldsymbol{X}\phi - \boldsymbol{v}_\Delta\|_{\boldsymbol{W}}^2 + \lambda(\mathbf{1}^T \phi - v_{all}), \tag{22}$$

where $\boldsymbol{v}_D = \boldsymbol{v} - v(\emptyset)\mathbf{1} \in \mathbb{R}^n$, $v_{all} = v(\boldsymbol{N}) - v(\emptyset) \in R$. Following the proof above, differentiating Eq. (22) and setting the derivative to 0, we obtain:

$$\nabla_\phi L = 2\boldsymbol{X}^T \boldsymbol{W}(\boldsymbol{X}\phi - \boldsymbol{v}_D) + \lambda \mathbf{1} = 0 \tag{23}$$

$$\nabla_\lambda L = \mathbf{1}^T \phi - v_{all} = 0 \tag{24}$$

After multiplying both sides of Eq. (23) we can get:

$$2\mathbf{1}^T \boldsymbol{X}^T \boldsymbol{W} \boldsymbol{X} \phi - 2\mathbf{1}^T \boldsymbol{X}^T \boldsymbol{W} \boldsymbol{v}_D + \lambda \mathbf{1}^T \mathbf{1} = 0$$

$$2\mathbf{1}^T \left( \frac{d-1}{d} \boldsymbol{I} + B \boldsymbol{J} \right) \phi - 2\mathbf{1}^T \boldsymbol{X}^T \boldsymbol{W} \boldsymbol{v}_D + \lambda d = 0$$

$$2 \left( \frac{d-1}{d} + Bd \right) \mathbf{1}^T \phi - 2\mathbf{1}^T \boldsymbol{X}^T \boldsymbol{W} \boldsymbol{v}_D + \lambda d = 0$$

$$2 \left( \frac{d-1}{d} + Bd \right) v_{all} - 2\mathbf{1}^T \boldsymbol{X}^T \boldsymbol{W} \boldsymbol{v}_D + \lambda d = 0$$

$$\frac{2}{d} \left[ \mathbf{1}^T \boldsymbol{X}^T \boldsymbol{W} \boldsymbol{v}_D - \left( \frac{d-1}{d} + Bd \right) v_{all} \right] = \lambda. \tag{25}$$

Substituting Eq. (25) into Eq. (23) yields:

$$2\boldsymbol{X}^T \boldsymbol{W} \boldsymbol{X} \phi - 2\boldsymbol{X}^T \boldsymbol{W} \boldsymbol{v}_D + \frac{2}{d} \left[ \mathbf{1}^T \boldsymbol{X}^T \boldsymbol{W} \boldsymbol{v}_D - \left( \frac{d-1}{d} + Bd \right) v_{all} \right] \mathbf{1} = 0$$

$$\frac{d-1}{d} \phi + B v_{all} \mathbf{1} - \boldsymbol{X}^T \boldsymbol{W} \boldsymbol{v}_D + \frac{1}{d} \boldsymbol{J} \boldsymbol{X}^T \boldsymbol{W} \boldsymbol{v}_D - \left( \frac{d-1}{d^2} + B \right) v_{all} \mathbf{1} = 0$$

$$\frac{d-1}{d} \phi - \boldsymbol{X}^T \boldsymbol{W} \boldsymbol{v}_D + \frac{1}{d} \boldsymbol{J} \boldsymbol{X}^T \boldsymbol{W} \boldsymbol{v}_D - \frac{d-1}{d^2} v_{all} \mathbf{1} = 0$$

$$\phi = \frac{d\boldsymbol{I} - \boldsymbol{J}}{d-1} \boldsymbol{X}^T \boldsymbol{W} \boldsymbol{v} + \frac{v(\boldsymbol{N}) - v(\emptyset)}{d} \mathbf{1}. \tag{26}$$

where $\boldsymbol{X}^T \boldsymbol{W} \boldsymbol{v}$ can be expanded in a cumulative form as follows:

$$\boldsymbol{X}^T \boldsymbol{W} \boldsymbol{v} = \sum_{\boldsymbol{S} \subsetneq \boldsymbol{N} \backslash \emptyset} \omega(\boldsymbol{S}) v(\boldsymbol{S}) \mathbf{1}^{\boldsymbol{S}} \approx \sum_{\boldsymbol{S} \sim \omega(\boldsymbol{S})} v(\boldsymbol{S}) \mathbf{1}^{\boldsymbol{S}}. \tag{27}$$

As can be seen, the least-square value is essentially equivalent to employing importance sampling with the Shapley kernel $\omega$.

## A.2 PROOF OF THE UNBIASNESS OF EQ. (14)

Equation (14) is unbiased in a probabilistic sense, as shown below:

$$
\begin{aligned}
\mathbb{E}[\phi_{\boldsymbol{x}}] &= \boldsymbol{T} \sum_{\boldsymbol{S} \subsetneq \boldsymbol{N} \setminus \emptyset} p^{ls}(\boldsymbol{S}) a_{\boldsymbol{S}} v(\boldsymbol{S}) + \boldsymbol{b} \\
&= \sum_{\boldsymbol{S} \subsetneq \boldsymbol{N} \setminus \emptyset} \frac{[(d - |\boldsymbol{S}|)\mathbb{I}_{i \in \boldsymbol{S}} - |\boldsymbol{S}|\mathbb{I}_{i \notin \boldsymbol{S}}]v(\boldsymbol{S})}{\binom{d}{|\boldsymbol{S}|}(d - |\boldsymbol{S}|)|\boldsymbol{S}|} + \frac{v(\boldsymbol{N}) - v(\emptyset)}{d}\boldsymbol{1} \\
&= \sum_{\boldsymbol{S} \subsetneq \boldsymbol{N} \setminus \emptyset, i \in \boldsymbol{S}} \frac{(d - |\boldsymbol{S}|)v(\boldsymbol{S})}{\binom{d}{|\boldsymbol{S}|}(d - |\boldsymbol{S}|)|\boldsymbol{S}|} - \sum_{\boldsymbol{S} \subsetneq \boldsymbol{N} \setminus \emptyset, i \notin \boldsymbol{S}} \frac{|\boldsymbol{S}|v(\boldsymbol{S})}{\binom{d}{|\boldsymbol{S}|}(d - |\boldsymbol{S}|)|\boldsymbol{S}|} + \frac{v(\boldsymbol{N}) - v(\emptyset)}{d}\boldsymbol{1} \\
&= \sum_{\boldsymbol{S} \subsetneq \boldsymbol{N} \setminus \emptyset, i \in \boldsymbol{S}} \frac{v(\boldsymbol{S})}{\binom{d}{|\boldsymbol{S}|}|\boldsymbol{S}|} - \sum_{\boldsymbol{S} \subsetneq \boldsymbol{N} \setminus \emptyset, i \notin \boldsymbol{S}} \frac{v(\boldsymbol{S})}{\binom{d}{|\boldsymbol{S}|}(d - |\boldsymbol{S}|)} + \frac{v(\boldsymbol{N}) - v(\emptyset)}{d}\boldsymbol{1} \\
&= \sum_{\boldsymbol{S} \subset \boldsymbol{N} \setminus \{i\}} \frac{v(\boldsymbol{S} \cup \{i\}) - v(\boldsymbol{S})}{\binom{d}{|\boldsymbol{S}|}|\boldsymbol{S}|(d - |\boldsymbol{S}|)}
\end{aligned}
$$

## A.3 PROOF OF THE EQUIVALENCE OF FASTSHAP IN EQ. (13)

We simply expand the original equation as follows ($g$ represents $g(\boldsymbol{x}; \theta)$ for short):

$$
\begin{aligned}
\mathcal{L} &= \mathop{\mathbb{E}}_{\boldsymbol{x} \in \mathcal{X}} \left[ \left\| g - (\boldsymbol{X}^T \boldsymbol{W} \boldsymbol{X})^{-1} \boldsymbol{X}^T \boldsymbol{W} \boldsymbol{v} \right\|_{\boldsymbol{X}^T \boldsymbol{W} \boldsymbol{X}}^2 \right] \\
&= \mathop{\mathbb{E}}_{\boldsymbol{x} \in \mathcal{X}} \left[ (g - (\boldsymbol{X}^T \boldsymbol{W} \boldsymbol{X})^{-1} \boldsymbol{X}^T \boldsymbol{W} \boldsymbol{v})^T \boldsymbol{X}^T \boldsymbol{W} \boldsymbol{X} (g - (\boldsymbol{X}^T \boldsymbol{W} \boldsymbol{X})^{-1} \boldsymbol{X}^T \boldsymbol{W} \boldsymbol{v}) \right] \\
&= \mathop{\mathbb{E}}_{\boldsymbol{x} \in \mathcal{X}} \left[ g^T \boldsymbol{X}^T \boldsymbol{W} \boldsymbol{X} g - 2 g^T \boldsymbol{X}^T \boldsymbol{W} \boldsymbol{v} + \boldsymbol{v}^T \boldsymbol{W} \boldsymbol{X} (\boldsymbol{X}^T \boldsymbol{W} \boldsymbol{X})^{-1} \boldsymbol{X}^T \boldsymbol{W} \boldsymbol{v} \right] \\
&= \mathop{\mathbb{E}}_{\boldsymbol{x} \in \mathcal{X}} \left[ \|\boldsymbol{X} g - \boldsymbol{v}\|_{\boldsymbol{W}}^2 \right] + \underbrace{\mathop{\mathbb{E}}_{\boldsymbol{x} \in \mathcal{X}} \left[ -\boldsymbol{v}^T \boldsymbol{W} \boldsymbol{v} + \boldsymbol{v}^T \boldsymbol{W} \boldsymbol{X} (\boldsymbol{X}^T \boldsymbol{W} \boldsymbol{X})^{-1} \boldsymbol{X}^T \boldsymbol{W} \boldsymbol{v} \right]}_{\text{Replace with constant } C} \\
&= \mathop{\mathbb{E}}_{\boldsymbol{x} \in \mathcal{X}} \left[ \|\boldsymbol{X} g - \boldsymbol{v}\|_{\boldsymbol{W}}^2 \right] + C.
\end{aligned} \tag{28}
$$

In summary, FastSHAP can be viewed as a special case of the unified amortized estimator proposed in this paper.

## A.4 ANALYSIS OF THE POTENTIAL NOISE OF $\phi_x$ IN EQ. (12)

We formulate this issue that for each input x, let the target be $\phi_x = \phi^* + \boldsymbol{n}$, where the first term is the true Shapley value and the second is an input-agnostic noise term. The noise term is independent of the input data (i.e., $p(\boldsymbol{x}, \boldsymbol{n}) = p(\boldsymbol{x})p(\boldsymbol{n})$) and the estimation is unbiased (i.e., $\mathbb{E}[\phi_x] = \phi^*$). We can decompose the loss function in Eq. (12) as follows:

$$
\begin{aligned}
L(\theta) &= \mathbb{E} \left[ \|g(\boldsymbol{x}; \theta) - \phi_{\boldsymbol{x}}\|_{\boldsymbol{M}}^2 \right] \\
&= \mathbb{E} \left[ \|g(\boldsymbol{x}; \theta) - \phi^* - \boldsymbol{n}\|_{\boldsymbol{M}}^2 \right] \\
&= \mathbb{E} \left[ \|g(\boldsymbol{x}; \theta) - \phi^*\|_{\boldsymbol{M}}^2 - 2 (g(\boldsymbol{x}; \theta) - \phi^*)^T \boldsymbol{M} \boldsymbol{n} + \|\boldsymbol{n}\|_{\boldsymbol{M}}^2 \right] \\
&= \mathbb{E} \left[ \|g(\boldsymbol{x}; \theta) - \phi^*\|_{\boldsymbol{M}}^2 \right] - \underbrace{\mathbb{E} \left[ 2 (g(\boldsymbol{x}; \theta) - \phi^*)^T \right] \boldsymbol{M} \mathbb{E}[\boldsymbol{n}]}_{\text{Unbiased estimation } \mathbb{E}[\boldsymbol{n}] = 0} + \mathbb{E} \left[ \|\boldsymbol{n}\|_{\boldsymbol{M}}^2 \right] \\
&= \mathbb{E} \left[ \|g(\boldsymbol{x}; \theta) - \phi^*\|_{\boldsymbol{M}}^2 \right] + \mathbb{E} \left[ \|\boldsymbol{n}\|_{\boldsymbol{M}}^2 \right] \\
&= \mathbb{E} \left[ \|g(\boldsymbol{x}; \theta) - \phi^*\|_{\boldsymbol{M}}^2 \right] + C \\
&= \text{Error}(\theta) + C,
\end{aligned} \tag{29}
$$

where $C = \mathbb{E}\left[\|n\|_M^2\right]$ and the metric matrix $M$ is positive definite. In other words, minimizing $L(\theta)$ is equivalent to simply minimizing the Shapley value estimation error: mean-zero noise in the target doesn't affect the optimum, because it becomes a constant in the objective function.

## A.5 SIMSHAP MODELS AND HYPERPARAMETERS

**Tabular Datasets.** For the original model $f(x, \eta)$, we adopted LIGHTGBM (Ke et al., 2017) for implementation. The surrogate model is implemented using 3 to 6 layer MLPs with 128/512 hidden units and ELU activations. The SimSHAP explainer model is implemented using neural networks that consist of 3 fully connected layers with 128/512 units and ReLU activations, according to the input feature of the dataset. It is trained AdamW optimizer with learning rate of range $7 \times 10^{-4}$ to $1.5 \times 10^{-2}$ and batch size of 1024/2048. All the models don't contain Softmax Layers. All the baselines for SimSHAP accuracy comparison were computed using an open-source implementation [4]. and fastshap package[5]. All the experiments were run on a GEFORCE RTX 3090 Card.

**Image Datasets.** The original model and surrogate model are both ResNet-18 structures. Specifically, the surrogate model takes 2 inputs, image and mask, and returns output of 10 dimensions. The SimSHAP explainer model is implemented using U-Net strucuture, with the output of $BatchSize \times y \times 16 \times 16$. It is trained using AdamW optimizer with learning rate of $2 \times 10^{-4}$ and batch size of 256. All the models don't contain Softmax Layers. To ensure comparability of all the methods, we average-pooled the results of methods that provide attributions of the same size as the image. Integrated-Gradients and SmoothGrad are implemented through `captum` package[6], KernelSHAP, KernelSHAP-S, and DeepSHAP are implemented through `shap` package[7], and Grad-CAM is implemented through an open-source package for explanation benchmarks[8].

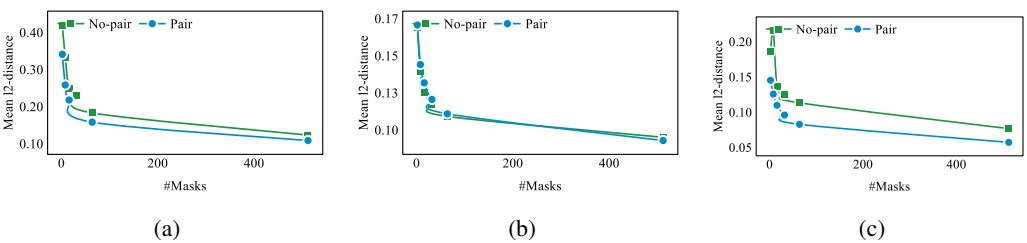

|     |     |     |
| --- | --- | --- |
| (a) | (b) | (c) |

Figure 5: SimSHAP accuracy as a function of number of training samples with/out pair sampling in (a) Census (b) News (c) Bank dataset.

Fig. 5 shows SimSHAP accuracy as a function of the number of training samples. Results reveals that across all tabular datasets, increasing the number of masks improves accuracy. However, as the number of masks increases, the accuracy gain becomes less significant. Additionally, pair sampling at least doesn't make the accuracy worse. In fact, pair sampling can improve accuracy in most cases.

## A.6 ABLATION STUDIES

**Tabular Dataset** We delve into an ablation study focusing on hyperparameters on `bankruptcy` dataset. We use the default configuration of 32 samples, pair sampling, and 1000 epochs for this study. Tables 5 to 8 presents the results. For learning rate, we observed that the best performance is achieved with learning rates ranging from $1 \times 10^{-3}$ and $1 \times 10^{-5}$ for 1000-epoch itereations. A high learning rate can lead to an unstable training process, while a low learning rate can lead to slow convergence. Therefore, in order to achieve the $O(\frac{1}{\sqrt{M}})$ convergence, learning rate should be chosen carefully. As for batch size, we find that larger batch sizes result in improved performance. As a

---

[4]https://github.com/iancovert/shapley-regression/ (License: MIT)

[5]https://github.com/iancovert/fastshap(License: MIT)

[6]https://captum.ai/docs/introduction/ (License: MIT)

[7]https://shap.readthedocs.io/en/latest/ (License: MIT)

[8]https://github.com/zbr17/ExplainAttr/ (License: MIT)

Table 5: Influence of learning rate.

|  | Mean $l_1$ distance | Mean $l_2$ distance |
|---|---|---|
| $1 \times 10^{-1}$ | 1.587 | 0.197 |
| $1 \times 10^{-2}$ | 2.051 | 0.224 |
| $1 \times 10^{-3}$ | 0.926 | 0.106 |
| $7 \times 10^{-4}$ | **0.886** | **0.098** |
| $1 \times 10^{-5}$ | 2.084 | 0.194 |
| $1 \times 10^{-6}$ | 5.950 | 0.548 |

Table 6: Influence of optimization epoch.

|  | Mean $l_1$ distance | Mean $l_2$ distance |
|---|---|---|
| 100 | 1.550 | 0.152 |
| 500 | 1.009 | 0.107 |
| 1000 | 0.886 | 0.098 |
| 1500 | **0.824** | **0.094** |

Table 7: Influence of batch size.

|  | Mean $l_1$ distance | Mean $l_2$ distance |
|---|---|---|
| 32 | 1.430 | 0.164 |
| 128 | 1.176 | 0.129 |
| 512 | 0.933 | 0.103 |
| 1024 | 0.886 | 0.098 |
| Whole Set | **0.857** | **0.092** |

Table 8: Influence of net structure

|  | Mean $l_1$ distance | Mean $l_2$ distance |
|---|---|---|
| wider | 0.933 | 0.108 |
| deeper | **0.865** | **0.099** |

result, the batch size can be increased as long as it remains within the constraints of available GPU memory. We evaluated performance across 100, 500, 1000, 1500 iterations. While performance improves with higher iteration counts, gains become less significant after 500 epochs. Striking a balance between accuracy and training time, we chose 1000 iterations for this dataset. We also explored the architecture of the explainer model. For MLPs, increasing the number of layers for a deeper model and increasing the hidden dimension for a wider model were considered. Results indicate that a 3-layer MLP with a hidden dimension of 512 is sufficient in capturing the Shapley space of raw data.

**Image Dataset** Similarly, we explored the hyperparameters of the **CIFAR-10** dataset. We use the default configuration of 8 samples, pair sampling, learning rate of $2 \times 10^{-4}$, and 500 epochs for this study. We first utilize **CIFAR-10** dataset to evaluate the performance of SimSHAP via the size of dataset. Results in Table 12 show the robustness of SimSHAP for achieving great performance with a mere 20% of training data. When more data is involved, SimSHAP can have small improvements but contribute to lower variance regarding to the AUC.

Detailed comparison are in Tables 9 to 11. For learning rate, we observe that the best performance is achieved with learning rates around $1 \times 10^{-4}$. For number of epochs, we find that 500-600 epochs is sufficient to balance the training speed and accuracy. More epochs may lower the variance for Inclusion and Deletion scores, but the improvement is not significant. For number of samples, we only tested the pair sampling cases. Results reveal that when the number of samples exceeds 8, the improvement is not significant, and there is a slight decrease in performance. This laid a funndation for a relatively fast and accurate training method with small number of samples on image datasets.

Table 9: Mean Insertion AUC and Deletion AUC of SimSHAP as a function of learning rate.

|  | Ins. AUC | Del. AUC |
|---|---|---|
| $1 \times 10^{-2}$ | 0.667 | **-0.365** |
| $1 \times 10^{-3}$ | 0.733 | -0.301 |
| $2 \times 10^{-4}$ | **0.755** | -0.302 |
| $1 \times 10^{-5}$ | 0.691 | -0.302 |
| $1 \times 10^{-6}$ | 0.498 | -0.188 |

Table 10: Mean Insertion AUC and Deletion AUC of SimSHAP as a function of limited data.

|  | Ins. AUC | Del. AUC |
|---|---|---|
| 200 | 0.726 | -0.332 |
| 400 | 0.740 | -0.312 |
| 600 | **0.777** | -0.288 |
| 800 | 0.720 | -0.302 |
| 1000 | 0.703 | **-0.348** |

Table 11: Mean Insertion AUC and Deletion AUC of SimSHAP as a function of batch size.

|     | Ins. AUC | Del. AUC |
| --- | --- | --- |
| 8   | 0.721 | -0.335 |
| 16  | 0.685 | -0.277 |
| 32  | 0.703 | **-0.337** |
| 64  | 0.716 | -0.298 |
| 256 | **0.755** | -0.302 |

Table 12: Mean Insertion AUC and Deletion AUC of SimSHAP as a function of epoch.

| Percent(%) | Ins. AUC | Del. AUC |
| --- | --- | --- |
| 4   | 0.532 | -0.168 |
| 16  | 0.736 | **-0.359** |
| 24  | 0.695 | -0.357 |
| 40  | **0.767** | -0.298 |
| 48  | 0.749 | -0.325 |
| 64  | 0.760 | -0.337 |
| 80  | 0.702 | -0.323 |
| 100 | 0.755 | -0.302 |

For batch size, similar to tabular datasets, a larger batch size may lead to better performance. But the parameter also needs to be carefully chosen for the GPU memory constraints.

### A.7 TRAINING ON INEXACT LABELS ON IRIS DATASET

In this section, we demonstrate SimSHAP's ability of training on inexact labels by conducting experiments on the Iris dataset. It's crucial to note that SimSHAP bypasses the need of ground truth labels for training by estimating them using limited sample data, which is also true for FastSHAP's Least Squares value (Jethani et al., 2021). The size of the dataset should be sufficiently large, or else the model might overfit to the noise introduced during the estimation process.

Table 13 demonstrates the results of the mean $l_2$ distance between the model's output and the estimated ground truth label, as the dataset size increases. Each configuration was trained until convergence. Together with the result in Table 12, we can conclude that SimSHAP is indeed capable of learning from noisy data when provided with a sufficiently large dataset. When the dataset is small, there is a higher likelihood that the model will learn the noise rather than the true Shapley values.

Table 13: Mean $l_2$ distance as a function of data size

| Size of Dataset | Distance |
| --- | --- |
| 5   | 0.577 |
| 20  | 0.144 |
| 45  | 0.070 |
| 60  | 0.052 |
| 75  | 0.050 |
| 100 | 0.042 |
| 120 (whole) | 0.033 |

### A.8 AN EXACT ESTIMATOR FOR UNBIASED KERNELSHAP

Following Section 3.2 of the unbiased KernelSHAP estimation (Covert & Lee, 2021), we provide "an approximate solution to the exact problem" with the Lagrangian as follows:

$$
\begin{aligned}
L(\eta, \lambda) = {} & \eta^T \mathbb{E}\left[(\mathbf{1}^{\boldsymbol{S}})(\mathbf{1}^{\boldsymbol{S}})^T\right] \eta \\
& - 2\eta^T \mathbb{E}\left[\mathbf{1}^{\boldsymbol{S}}(v(\boldsymbol{S}) - v(\emptyset))\right] \\
& + \mathbb{E}\left[(v(\boldsymbol{S}) - v(\emptyset))^2\right] \\
& + \lambda(\mathbf{1}^T \eta - v(\boldsymbol{N}) + v(\emptyset)).
\end{aligned}
\tag{30}
$$

Using the shorthand notation

$$
A = \mathbb{E}\left[(\mathbf{1}^{\boldsymbol{S}})(\mathbf{1}^{\boldsymbol{S}})^T\right], \;\; b = \mathbb{E}\left[\mathbf{1}^{\boldsymbol{S}}(v(\boldsymbol{S}) - v(\emptyset))\right],
$$

we can calculate $A$ precisely and only need to estimate $b$ by Monte Carlo Sampling:

$$\bar{b}_M = \frac{1}{M}\sum_{k=1}^{M}\mathbf{1}^{\boldsymbol{S_k}}v(\boldsymbol{S_k}) - \mathbb{E}\left[\mathbf{1}^{\boldsymbol{S}}\right]v(\emptyset). \tag{31}$$

The unbiased KernelSHAP is formulated as follows:

$$\eta_M = A^{-1}(\bar{b}_M - \mathbf{1}\frac{\mathbf{1}^T A^{-1}\bar{b}_M - v(\boldsymbol{N}) + v(\emptyset)}{\mathbf{1}^T A^{-1}\mathbf{1}}). \tag{32}$$

We recommend readers to refer to Section 1 of the Supplementary material in Covert & Lee (2021).

### A.9    DETAILED COMPARISON BETWEEN OUR WORK AND SCHWARZENBERG ET AL. (2021); CHUANG ET AL. (2023)

**About the Similarity**    We acknowledge the similarities between our work and Schwarzenberg et al. (2021); Chuang et al. (2023), but it should be noted that both works not only deal with Shapley values computation. Firstly, Schwarzenberg et al. (2021) proposed a general framework based on the concept of amortized estimation, in which a single neural network is trained to predict the Integrated Gradients / Shapley values for each player (either a pixel in an image or a token in text). Secondly, Chuang et al. (2023) is rooted in RTX, which is fundamentally equivalent to amortized estimation (with FastSHAP mentioned as the first line of work or the RTX paradigm).

**About the Difference**    However, there are also notable distinctions. Schwarzenberg et al ensures accuracy in the first term of their Eq.1: $argmin_{\theta\in\Theta}\frac{1}{|X|}\sum_{x\in\mathbf{X}}\alpha D(E_f(x), e_\theta(x)) + \beta(\frac{\|e_\theta(x)\|}{\|E_f(x)\|})$. If we were to utilize ground truth labels, our framework would align with this approach, matching the $E_f(x)$ of the above equation, indicating the expensive explainer. Nevertheless, we didn't apply any form of supervision, instead relying solely on sampled ground truth data for our MSE computations. For Chuang et al. (2023), it is mentioned in the Introduction section of CoRTX that methods like Fastshap "learn an explainer to minimize the estimation error regarding to the approximated explanation labels". We concur with this assessment, though this work does not provide a definitive explanation on this matter. FastSHAP, based on Eq. (13), represents a specific example of approximation with a complicated metric matrix, and these 2 works can also be seen as specific examples of our proposed framework. Furthermore, both the fine-tuning stage of CoRTX and the Supervised RTX baseline they used require ground truth labels for training, which is not the case in SimSHAP. Similarly to FastSHAP, we do not acquire any ground truth labels during training.

### A.10    ANALYSIS OF THE TIME COMPLEXITY OF FASTSHAP AND SIMSHAP ALGORITHMS

In addition to the unified framework, we also want to highlight that the time complexity of SimSHAP and FastSHAP is comparable, which corresponds to the amortized estimation version of the semi-value and least squares value.

We define $B$ as the number of batches, $d$ as the input feature dimension, $K$ as the number of samples, and $o$ as the output dimension of the classifier (which refers to number of classes).

We can rewrite the main loss function of FastSHAP as follows:

$$L_{fastshap} = \sum_{b=1}^{B}\sum_{y=1}^{o}\sum_{k=1}^{K}(v_y(\boldsymbol{S_k}) - v_y(\emptyset) - \boldsymbol{S_k}^T\hat{\phi})^2,$$

where $v_y(\cdot)$ indicates the yth component of vector $v(\cdot)$. Ignoring the computation cost of $v(\boldsymbol{S_k}) - v(\emptyset)$, the number of multiplications is $(d+2)KoB$, and the number of additions is $dKoB$.

Similarly, we can rewrite the main loss function of SimSHAP as follows:

$$L_{simshap} = \sum_{b=1}^{B}(\hat{\phi} - \phi_{sample})^2, \tag{33}$$

where

$$\phi_{sample} = \sum_{k=1}^{K} \omega(\boldsymbol{S}_k) v^T(\boldsymbol{S}_k) + \frac{v(\boldsymbol{N}) - v(\emptyset)}{d}. \tag{34}$$

Note that $\hat{\phi}$ has the dimension of $o \times d$. Ignoring the computation cost of $\frac{v(\boldsymbol{N})-v(\emptyset)}{d}$, the number of multiplications is $(dKo+1)B$, the number of additions is $(2do + K - 1)B$.

According to the above comparison, the time complexities of the tow algorithms are $\mathcal{O}(dKoB)$ under the framework of Big O notation.

