# OpenReview forum: "Exploring Unified Perspective For Fast Shapley Value Estimation"
_ICLR.cc/2024/Conference — Submitted to ICLR 2024_

### Official Review · Reviewer_UWrD · 2023-10-31

**Soundness:** 2 fair
**Presentation:** 1 poor
**Contribution:** 2 fair
**Rating:** 3
**Confidence:** 4

**Summary:**

This work considers the topic of efficient Shapley value estimation and presents two contributions. The first is that several current Shapley value estimators can be understood via a shared perspective of a linear transformation of expectations of feature subsets. The second is that amortized Shapley value estimation can be viewed via comparison to the true Shapley values in a chosen metric space, which enables the development of a new approach, SimSHAP. Under this method, we simply calculate the similarity to estimated Shapley values, and this is competitive with an existing amortized method (FastSHAP).

**Strengths:**

Efficient Shapley value estimation is an interesting topic that is in need of improved methods, particularly ones that leverage amortization to provide faster solutions than stochastic estimators. This work provides a new perspective on existing stochastic estimators, although it does appear to derive any practical insights or algorithms from this perspective. It then provides another new perspective on amortized methods, and the authors use this to propose SimSHAP. SimSHAP is simple and competitive with FastSHAP.

**Weaknesses:**

About the main methodological contributions:

- The main contribution of this paper is SimSHAP, which involves training a Shapley value estimation model by comparing to estimated values for each gradient step. While there is a lot of build-up to this proposal, this idea has been explored by at least two other works [1, 2], so the novelty is limited. These works are not cited, and I'm not sure there's much methodological innovation on top of them.

- Regarding the unified perspective in Section 2.3, it seems like this view is clearly apparent for the semivalue-based estimator and does not add much here. After reading the paper closely, I'm not sure it's correct for the least squares estimator. Previous work by Covert & Lee (2021) showed a similar derivation from the KKT conditions, and the analytic solution involves two terms that are estimated: one that resembles the importance sampling term, and one that resembles the transform term (see their section 3.2). In this work, it looks like the transform term $T$ is assumed to be known, which does not correspond to KernelSHAP but rather an alternative version from that work called "unbiased KernelSHAP." Are the authors perhaps analyzing that version rather than the original one? Note that the original one cannot be calculated without a sufficient number of samples due to invertibility issues (the authors here manage to use a very small number in their experiments, suggesting the unbiased version), and that the original one is not provably unbiased (whereas the authors claim zero bias here in eq. 14).

- Regarding the unified view of amortized estimators, it is true that you can compare to the true Shapley values with any properly defined metric and learn the correct function. It is therefore not surprising that you can use the l2 distance. The authors might have been more careful about showing why it's okay to train on inexact values from the least squares estimator, because these training targets will be very noisy. What is the impact of this noise during training? Also, note again that [1, 2] have already explored this approach.

- Regarding the unified view of amortized estimators, I'm not sure the proposed view of FastSHAP is correct. For the unnumbered equation defining $\mathcal{L}(\theta)$ in Section 2.3, there are two equations and I'm not able to see whether they are mathematically equivalent. The authors should consider adding this derivation, either here or in the appendix. Furthermore, the expression for $\phi_x$ is not equal to the Shapley values, because it isn't the solution from the KKT conditions and doesn't incorporate the constraint. Currently, I do not see how this perspective is correct, and it doesn't seem helpful to view the FastSHAP loss as comparing to the true values under a complicated metric: it misses the main point of the FastSHAP loss, which is that it bypasses the need for the ground truth. This perspective is also not necessary to understand that comparing to the ground truth using l2 distance should work.

- Related to the above point, I'm not sure the entries for FastSHAP in Table 2 are correct.

[1] Schwarzenberg et al, "Efficient Explanations from Empirical Explainers" (2021)

[2] Chuang et al, "CoRTX: Contrastive Framework for Real-time Explanation" (2023)

About the experiments:

- In Figure 2, SimSHAP does not appear to outperform FastSHAP. In fact it's noticeably worse for the census dataset. Why is that, and can it be made more accurate?

- The results in Table 3 appear to contradict those in Figure 4. How can SimSHAP have the best AUC when its curves are far from the best?

About the writing, which could be significantly improved:

- The description of related work is incorrect in several places. In the introduction, "semivalue" and "least squares value" are not estimation methods, they are classes of game-theoretic solution concepts, and their mathematical perspectives are used to develop estimation methods (e.g., fitting a weighted least squares model with an approximate version of the least squares objective). Also in the introduction, model-specific methods do not "reduce computation cost by merging or pruning redundant feature combinations," this is not true for TreeSHAP, LinearSHAP or DeepSHAP. It is also not true that "the exact differences among these algorithms remain unclear," the differences seem very clear, as they are described in this work. The gap in missing work seems to be that the fastest solutions either are not general enough (TreeSHAP) or accurate enough (FastSHAP), although this work does not appear to solve the latter problem.

- The "large numeral law" is not common terminology, it may be better to call this the "law of large numbers" as it is typically called (see [wikipedia](https://en.wikipedia.org/wiki/Law_of_large_numbers)). Also, the LLN does not imply an approximation error of $\mathcal{O}(1/\sqrt{M})$, it only implies convergence of the sample mean to the expectation. For the error rate, it might be better to cite a probabilistic error bound like Chebyshev's inequality.

- For proposition 1, it would be helpful to refer to a proof. Or even better, because this is a known result, the authors might refer to one of the works that originally showed it to be true, for example Charnes et al 1988.

- In eq. 8 regarding FastSHAP, the authors' notation neglects to show the input $x$ in the cooperative game $v(S)$. It may be difficult for readers to follow what $v(S)$ represents. It would also help to give a specific example of how $v(S)$ is defined in Section 2.1, because no equation is currently provided.

- Regarding the additive efficient normalization, there are parts where FastSHAP is described as "biased" or that its predictions need to be rectified. The current description could be confusing to many readers, because it suggests that the loss is somehow incorrect, but that's not quite true. For example, you could write that the weighted least squares loss does not encourage the correct optimal value unless the predictions satisfy the efficiency constraint, which can be enforced by applying the normalization to the predictions.

- There are small typos throughout the paper, for example "remarkbly" on page 3 and "minimizaion" on page 4. The paper could be more thoroughly proofread.

- Overall, the paper's structure is extremely similar to the FastSHAP paper, to the point that the authors have replicated all the main text experiments, many of the supplementary experiments, and have nearly copied some of the writing. It would be helpful to either acknowledge this similarity, or deviate from it more significantly.

**Questions:**

Several questions are mentioned in the weaknesses section.

---

> ### Author Response · Authors · 2023-11-20
> **Response to Reviewer UWrD (part 1/2)**
>
> We thank reviewer UWrD for all the helpful critiques and suggestions regarding our work. We have made targeted revisions and additions.
> For the revisions of the paper, we have marked them in **green** in the PDF.
>
> ## About the main methodological contributions
>
> **W1:**
>
> Thanks for reviewer's suggestions, we have added two references to the revised version.
> - **About missing references:** We also noticed these works in our methodological survey, and we acknowledge that there have been some research efforts on amortized estimators in recent years, such as the FastSHAP method cited in our paper.
> - **About contributions:** We apologize for the unclear parts in the paper, but we still want to emphasize the main contributions of this paper.
> Many recent works have studied different estimation methods for approximating Shapley values, such as **semi-value** (ordinary Monte Carlo estimation) and **least-square value** (using the weighted least squares property of Shapley values).
> However, we found that these methods are **essentially not different**.
> We find that the least-square value is essentially still equivalent to some special probability Monte Carlo estimator.
> Therefore, we propose a unified perspective of stochastic estimators and build a unified perspective of amortized estimators on this basis.
> We have emphasized the main contributions of the paper in the revised version.
>
> **W2:**
>
> Sorry for the confusion in the original paper.
> - **Least-square value & unified perspective:** Regarding whether the least-square value can be incorporated into our unified perspective, we have provided detailed analysis and rigorous proof in **Appendix A.1**.
> - **About unbiased kernelSHAP:** Thanks to the reviewer for pointing out that we did use the **unbiased kernelSHAP** in this work, and we have made it clearer in the revised version to eliminate any possible confusion.
> - **Proof of Eq(14):** Regarding the proof of the unbiasedness of the original formula 14, we have provided detailed derivations in **Appendix A.2** and ensured its unbiasedness.
>
> **W3:**
>
> Thanks for the suggestion.
> - **About the fitting priority of information with different frequency:** We acknowledge the need to clarify how the model can be trained on noisy data. We have added the analysis in **Appendix A.4**.
> - **Supplementary exprimental results:** The model can ensure a fit to the true value if the training set is large enough.
> As the reviewer pointed out, a small dataset can cause the model to overfit the noise and fail to capture the true value. We plan to research ways to learn the true value with limited data as our future direction. We have demonstrated the above statement with experiments in **Appendix A.7**. Furthermore, our sampling strategy guarantees convergence to the ground truth value when operating on sufficient amount of data. Together with Table 12 in **Appendix A.6**, we can ensure the feasibility of our algorithm.
>
> **W4:**
>
> - **About the theoretical derivation of Eq(13):** We are sorry to have deleted the theoretical derivation due to the page limit. We can ensure its rightness and have included the whole proof in **Appendix A.3**.
> - **About $\phi_x$'s analysis:** In addition, we admit the inaccuracy of $\phi_x$ to Shapley values. That's why we add the $\frac{v(N) - v(\phi)}{d}\textbf{1}$ term to make it strictly equal to Shapley values, which is also a technical difference between **SimSHAP** and **FastSHAP**.
>   - **For FastSHAP:** FastSHAP utilizes normalization/penalty punishment terms in the loss to ensure correctness.
>   - **For SimSHAP:** We have found that the issue can be resolved by simply incorporating this term into the calculation of the sampled ground truth, allowing for the construction of SimSHAP without the need for normalization or penalty in the end.
> - **About the main point of FastSHAP:** We acknowledge its contribution by bypassing the need for the ground truth labels. Here we would like to reiterate our contribution in unifying and proposing SimSHAP:
>   - for stochastic estimator (Table 1), we demonstrated that the least squares value with KKT condition is equivalent to direct Monte-Carlo sampling on the classic form of Shapley values computation. This serves as the answer to why we didn't discover a significant difference in computational complexity (see **Appendix A.1**).
>   - What's more, We would emphasize that SimSHAP bypasses the need for ground truth Shapley values labels as well, but we choose the simplest metric and function as a result of our unified framework.
>
> **W5:**
>
> We have addressed the questions above, primarily on W4.

---

> ### Author Response · Authors · 2023-11-20
> **Response to Reviewer UWrD (part 2/2)**
>
> ## About the experiments
>
> **Q1:**
>
> Thanks for the insightful suggestion.
> - The results in **Appendix A.6-7** demonstrate that significantly enlarging the sample size can notably enhance the model's fitting performance, but it also introduces additional computational burdens.
> - To maintain consistent training times compared to the baseline approach, the results depicted in **Figure 2** do not incorporate a large sample size. Future research endeavors will focus on exploring how to achieve superior fitting outcomes with smaller sample sizes, as outlined in **W3 in part 1**.
>
> ## About the writing
>
> We appreciate the reviewer's attention to detail and comprehensive understanding of our paper.
>
> **Q1:**
>
> We admit our incorrectness in the "Related Work" section and "Introduction" section.
> - Regarding terms such as semivalue, we follow the terminology used in [1].
> - We have revised the paper and modified "reduce computation cost by merging or pruning redundant feature combinations" to "introduce model-specific information" in order to be more appropriate and accurate.
> - Moreover, we have to clarify "the exact differences among these algorithms remain unclear". Our motivation is that there are so many model-agnostic methods for Shapley values estimation and we sometimes don't know which method to choose for a specific scenario, so we turned to explore their differences in essence. We thus articulated this motivation in the new version of our paper.
>
> **Q2:**
>
> We apologize for not double-checking the official term for "law of large numbers". In addition, We are referring to the convergence of the sample mean to the expectation by mentioning $O(\frac{1}{\sqrt{M}})$, and we think the reviewer's suggestion makes the statement more precise.
>
> **Q3:**
>
> We have proofed **proposition 1** in **Appendix A.1**, and we also accept the advice to emphasize the result by citing the work the reviewer mentioned.
>
> **Q4:**
>
> We apologize for any confusion in the notation system.
> We have organized the symbols in the new version.
>
> **Q5:**
>
> The reviewer is right as we did not intend to demonstrate the FastSHAP loss as a biased loss, but it needs to be normalized which corresponds to satisfying the efficiency constraint. It's also worth mentioning that we added one term instead of performing normalization for Shapley values correction in our algorithm, as stated in **W4** for weakness.
>
> **Q6-7:**
>
> We acknowledge that we followed FastSHAP's experimental setup to provide a systematic evaluation of our method.
> We have clearly stated this in the revised version of the paper and have also corrected all the typos pointed out by the reviewer, which has contributed to a more organized and precise presentation.
>
> ### Reference:
>
> [1] Algorithms to estimate Shapley value feature attributions, Chen, Hugh and Covert, Ian C and Lundberg, Scott M and Lee, Su-In, Nature Machine Intelligence, 2023

---

> > ### Comment · Reviewer_UWrD · 2023-11-20
> > **Response (1/2)**
> >
> > Thanks to the authors for their response, and I think the new edits to the paper are helpful. I believe this work is close to being publication-ready, but there are still some significant issues:
> >
> > **Prior works.** It's not sufficient to simply cite Schwarzenberg et al and Chuang et al, what they propose is basically identical to SimSHAP: an explainer model is trained by penalizing the error relative to a common approximation algorithm. The paper should acknowledge this similarity and discuss differences, if the authors believe there are any. If there are not, I believe the main contribution of this paper is to provide a new perspective on FastSHAP being a Mahalanobis distance with a complex weighting. Personally, I don't think this perspective is important to motivate SimSHAP, given that it was independently proposed by these earlier works.
> >
> > **Similarity of semivalue and least squares estimators.** Thanks for correcting the work and acknowledging that it uses unbiased KernelSHAP rather than the original version. I still think it's confusing to call this the "least squares estimator" throughout the paper: readers will associate the least squares perspective with KernelSHAP rather than the unbiased version from Covert & Lee, because the latter converges slower and is rarely used (see that work's experiments). Also, the result about it being unbiased is known from that work, so it does not need to be proven again here (Appendix A.2). Finally, the perspective that it's equivalent to a complicated Monte Carlo estimator is unfortunately also known: see Section 4.2 of "SHAP-IQ: Unified Approximation of any-order Shapley Interactions."
> >
> > **FastSHAP's equivalence to eq. 13.** Thank you for including the proof in Appendix A.3, it's very helpful. However, there's one detail that isn't handled carefully: for the final expression to be equivalent to the FastSHAP loss, we require that $W$ contains $\infty$ for the entries corresponding to the empty and full sets. This detail is handled a bit casually, it's not even mentioned. Is every equality in this analysis valid when $W$ has these $\infty$-valued entries?
> >
> > **Noisy labels.** I believe the authors are correct that it's okay to fit with noisy labels, but I can't follow the current reasoning about "high-frequency information." This explanation is vague and very informal, but I think there's a simple explanation for why noisy labels are okay. If I may make a suggestion, I believe the rationale is the following:
> >
> > For each input $x$, let the target be $\phi_x = \phi(x) + \tilde \phi(x)$ where the first term is the true SV and the second is an input-specific noise term such that $\mathbb{E}[\tilde \phi(x)] = 0$. Define your loss function as $\mathcal{L}(\theta) = \mathbb{E}[ ||g(x; \theta) - \phi_x ||^2 ]$. If we decompose this, we can see that $\mathcal{L}(\theta) = \mathbb{E}[ ||g(x; \theta) - \phi(x) - \tilde \phi(x) ||^2 ] = \mathbb{E}[ ||g(x; \theta) - \phi(x)||^2 ] + C = \text{Error}(\theta) + C$, where $C = \mathbb{E} [||\tilde \phi(x)||^2 ] = \mathbb{E}_x [\text{Tr}(\text{Cov}(\tilde \phi(x)))]$. In other words, minimizing $\mathcal{L}(\theta)$ is equivalent to simply minimizing the Shapley value estimation error: mean-zero noise in the target doesn't affect the optimum, because it becomes a constant in the objective function.
> >
> > Zooming out, the implication is that by performing SGD on $\mathcal{L}(\theta)$, you're minimizing the Shapley value estimation error without having access to the ground truth. (The authors pointed this out in their response, but I think this proof makes the point more concrete.) This is interesting, and in addition to the Mahalanobis perspective on FastSHAP, I wonder if the authors can contextualize this finding by comparing with Theorem 1 from "Learning to Estimate Shapley Values with Vision Transformers," which shows that minimizing the FastSHAP loss is equivalent to minimizing an upper bound on $\text{Error}(\theta)$. The current treatment of this topic is a bit shallow, and it could become a more interesting contribution in this work.
> >
> > **Definition 3.** I noticed that this definition mentions the possibility of using estimated SV's for $\phi_x$ rather than the true values. I believe it's worth repeating the analysis above for different Mahalanobis distances to verify that the objective's optimum is unchanged regardless of $M$ when $\phi_x$ is an unbiased estimate. If that's not true, the paper should probably mention that eq. 12 only allows the use of unbiased estimates when $M = I$.

---

> > > ### Comment · Reviewer_UWrD · 2023-11-20
> > > **Response (2/2)**
> > >
> > > **Appendix A.7 results.** The Iris experiment provides some empirical evidence that fitting with noisy labels is okay and improves with more data. But I would recommend repeating this experiment with a more realistic dataset.
> > >
> > > **Figure 4 vs. Table 3.** The authors' response did not address the inconsistency between these results. SimSHAP's results for the insertion curve in Figure 4 are much worse than KernelSHAP, so how can it have higher insertion AUC in Table 3?
> > >
> > > Other issues:
> > >
> > > - The notation $v(S) = f(1^S · x)$ is hardly a clarification, unfamiliar readers will not be able to infer what this means.
> > > - Even with the now-clarified role of unbiased KernelSHAP, eq. 5 is not correct. The second line of this equation, beginning with $\approx$, defines the original KernelSHAP rather than the unbiased version! It may be worth revisiting Covert & Lee to see the difference between "an exact solution to the approximate problem" and "an approximate solution to the exact problem." Eq. 5 shows the former, but unbiased KernelSHAP is the latter.
> > > - Regarding the law of large numbers, the correction is still wrong. The LLN does not imply the quoted rate of $O(1/\sqrt{M})$.
> > > - It might be worth clarifying how the authors suggest tuning parameters for SimSHAP. Given that we don't assume access to the ground truth, how should we perform model selection or early stopping?
> > >
> > > Overall, this work still has many issues and would benefit from more time to make revisions.

---

> ### Author Response · Authors · 2023-11-30
> **Response to Reviewer UWrD (part 1/2)**
>
> We thank reviewer UWrD for all the additional suggestions regarding our work. We have made targeted revisions and additions. For the revisions of the paper, we have marked them in **green** in the PDF.
>
> ## Major Issues
>
> **Prior Works:**
>
> - **About the similarity:** We acknowledge the similarities between our work and those of Schwarzenberg et al and Chuang et al, but it should be noted that both works not only deal with Shapley value computation. Firstly, Schwarzenberg et al proposed a general framework based on the concept of amortized estimation, in which a single neural network is trained to predict the IG / Shapley value for each player (either a pixel in an image or a token in text). Secondly, Chuang et al's work is rooted in RTX, which is fundamentally equivalent to amortized estimation (with FastSHAP mentioned as the first line of work or the RTX paradigm.)
> - **About the difference:** However, there are also notable distinctions. Schwarzenberg et al ensures accuracy in the first term of their Eq.1: $argmin_{\theta\in \Theta}\frac{1}{\left|X\right|}\sum_{x\in \textbf{X}}\alpha D(E_f(x), e_{\theta}(x)) + \beta(\frac{\left\|e_{\theta}(x)\right\|}{\left\|E_{f}(x)\right\|})$. If we were to utilize ground truth labels, our framework would align with this approach, matching the $E_f(x)$ of the above equation, indicating the expensive explainer. Nevertheless, we didn't apply any form of supervision, instead relying solely on sampled ground truth data for our MSE computations. For Chuang et al, it is mentioned in the Introduction section of CoRTX that methods like Fastshap "learn an explainer to minimize the estimation error regarding to the approximated explanation labels". We concur with this assessment, though CoRTX does not provide a definitive explanation on this matter. Here we would like to **emphasize our primary contribution**, which is to unify current frameworks of Shapley Values estimation. FastSHAP, based on our Eq. 13, represents a specific example of approximation with a complicated metric matrix, and Schwarzenberg et al and Chuang et al can also be seen as a specific example of our proposed framework.
> - **About the motivation for proposing SimSHAP:** We admit that the proposed SimSHAP is indeed insufficiently motivated. Our motivation stems from the fact that within a unified framework, we can select the semi-value equation and Identity matrix due to their simplicity. Additionally, we aim to identify the optimal metric matrix that maximizes performance in terms of optimization convergence as our future work.
>
> We have revised our paper according to the reviewer's suggestions.
>
> **Similarity of semivalue and least squares estimators:**
>
> - **About the comparable time complexity:** Besides the unified framework, we would also like to emphasize that the time complexity of SimSHAP and FastSHAP is comparable, which corresponds to the amortized estimation version of semi value and least squares value. We have provided detailed analysis in our Appendix A.10. We also want to clarify that our focus is on the **static properties** of the algorithms, rather than the dynamic optimization properties.
> - **About the naming of "least squares estimator":** We have carefully reviewed our paper and did not find the term "least squares estimator" anywhere in the text. Additionally, we mentioned the "least squares value" in Section 2.1 and explained that it is equivalent to the definition of the Shapley value, which it has nothing to do with the estimation strategies such as Monte Carlo Sampling discussed later in the paper.
>
> **FastSHAP's equivalence to eq. 13:**
>
> Thanks for the reviewer for pointing out the $\infty$ entries issue.
>
> In our paper, all references to W pertain to the non-empty proper subset of W, implying that W encompasses a total of $2^d-2$ dimensions. We have taken into account two special scenarios that might contain $\infty$ values (relevant symbol definitions can be found below Eq. 12 and in Appendix A.1)
>
> **Noisy Labels:**
>
> Thanks for the reviewer for providing detailed derivation about fiting with noisy labels. This proof is quite intricate and we have updated our approach with this more refined proof.
>
> - **About training with limited data:**  Since Mean Squared Error (MSE) is a strongly convex function, an upper bound can be derived analogously in the case of finite samples as Theorem 1 in the ViT paper [1].
> - **About comparison with [1]:** We admit that the current treatment of this topic is a bit shallow. In our revised paper, we have provided a complete analysis for training without ground truth labels. Additionally, as part of our future work, we aim to identify the optimal metric matrix that maximizes performance in terms of optimization convergence.

---

> > ### Author Response · Authors · 2023-11-30
> > **Response to Reviewer UWrD (part 2/2)**
> >
> > ## Major Issues
> >
> > **Issue for Definition 3:**
> >
> > - **About the objective's optimum under different metric matrices:** For different metric matrices to achieve an equivalent optimal objective, they must be positive definite; otherwise, additional regularization terms are necessary for correction.
> > - **About the approximation of $\phi_{x}$:** Thanks for the reviewer for the suggestion. We have identified the imprecision in our statement, and have addressed these issues in the revised version of our paper. For simplicity, our Eq. 12 focuses on the derivation of the case where $\phi_{x}$ represents an exact value.
> >
> > **About Appendix A.7 results:**
> >
> > Thanks for the reviewer for the suggestion. However, due to time constraints, we are unable to implement these modifications in this current rebuttal phase. We intend to address these concerns in subsequent revisions as part of our future work.
> >
> > - **About CIFAR 10 result on data size:** Additionally, we would like to draw attention to our previous CIFAR-10 experiments on different dataset sizes in Table 12. Through these two investigations, we can at least demonstrate that when the dataset is extremely small, it may result in overfitting of noise, which is in line with our theoretical derivations.
> >
> > **Figure 4 vs. Table 3:**
> >
> > We're double-checking the details of this experiment and sorry for the late reply. Previously, we normalize the score curve to begin with 0 for insertion (1 for deletion) and end with 1 for insertion (0 for deletion).  While computing the mean and standard deviation, we only satisfy that the network output for the **original image (grand coalition)** is the same. That's why inconsistency is posed. We only enforced the latter constraint now and we can ensure the consistency between Figure 4 and Table 3.
> >
> >
> >
> > ## Other Issues
> >
> > We appreciate the reviewer's attention to comprehensive understanding of our paper and detailed suggestions.
> >
> > **W1:**
> >
> > We have added the clarification right after $v(S)= f(1^S\cdot x)$ to clarify the notation and avoid any possible confusion.
> >
> > **W2:**
> >
> > The reviewer is right about the unbiased Kernelshap estimation. We have modified Eq.5 in our latest revision.
> >
> > **W3:**
> >
> > We admit it's not rigorous enough to derive from the law of large numbers to the convergence rate. We would like to clarifty that the Lindeberg-Levy Central Limit Theorem (CLT), the variance of the random variable is related to the quantity M and is of order $\mathcal{O}(1/\sqrt{M})$.
> >
> > **W4:**
> >
> > **About model selection:** For fairness in evaluation, we choose the same model as FastSHAP.
> >
> > **About Tuning Hyperparameters:** We employed grid search to find the best set of hyperparamters.
> >
> > **About Early Stopping:** We did not implement a carefully designed training procedure, so we did not utilize the technique of early stopping.
> >
> >
> > ### References:
> >
> > [1] Covert et al, "Learning to Estimate Shapley Values with Vision Transformers" (2022)

---

> > > ### Author Response · Authors · 2023-11-30
> > > **Look Forward to further Comments and Suggestions**
> > >
> > > Thanks for the insightful suggestions provided by the reviewer UWrD. We have given targeted responses to each concern and made relevant adjustments to the original paper. We are more than willing to receive further remarks and recommendations from the reviewer on our submission and responses.
> > >
> > > If the reviewer feels that our response has adequately addressed each concern, we respectfully request you to raise the rating of our paper. Thank you very much!

---

> > > > ### Comment · Reviewer_UWrD · 2023-12-01
> > > > **Response (1/2)**
> > > >
> > > > Thanks to the authors for their response and updates, it’s nice to see such commitment to improving the paper. This last round of changes is a significant improvement, but unfortunately I still see issues that need to be resolved. Sorry for being picky, but there are many places where this work can be more careful and precise. I’m confident the authors can manage this for a future submission given more time to revise it.
> > > >
> > > > Prior works:
> > > > - These are the most important related works to discuss, doesn't the appendix A.9 discussion belong in the main text? Mentioning in a footnote that there’s an appendix discussion almost hides it from the reader.
> > > > - The difference with Schwarzenberg et al. (2021) seems correct. Although this work also uses SVs as prediction targets, they use a different loss.
> > > > - CoRTX discusses training on Shapley values with MSE, which is very similar to SimSHAP: see line 8 of algorithm 1, and the supervised RTX baseline (which is basically SimSHAP). It also has some kind of pretraining setup, so SimSHAP-style training is basically one stage of CoRTX. I can’t understand what differences the authors are trying to point out in their last response. However, reading the CoRTX paper again there is perhaps one difference: CoRTX trains with a small amount of accurate labels (after pretraining), whereas SimSHAP trains with many inaccurate labels (closer to FastSHAP, which doesn’t technically use labels). In other words, the new derivation about training with noisy labels is a legitimate difference that can be highlighted.
> > > >
> > > > Estimators:
> > > > - Section 2.2.1 is still written strangely. Unbiased KernelSHAP is mentioned briefly but not shown. Then, after mentioning it, the authors say that $\mathcal{L}(\eta)$ needs to be solved approximately, and only then do they introduce KernelSHAP - why? Unbiased KernelSHAP is designed for the same purpose and was already mentioned, so it's just hard to follow.
> > > > - The remark about solving the $\mathcal{L}(\eta)$ KKT conditions is presented as a new result, and the authors point to a derivation in appendix A.1. But Covert & Lee shows this already, why not just mention it when citing that work?
> > > > - For clarity, shouldn’t the second row of Table 1 be labeled “Unbiased KernelSHAP” rather than “Least Squares Value?” The current name is unclear about which estimator the authors are referring to.
> > > > - Under the “Least Squares Value” part of Section 2.2.1, it might be worth specifying that this view on the Shapley value is used by unbiased KernelSHAP (it's currently not mentioned here). The work mentioned in my previous response, Fumagalli et al 2023, should probably also be mentioned here, as it already shows that the unbiased KernelSHAP solution is a linear transformation of a Monte Carlo estimator.
> > > > - The authors included an entry in Table 1 for a new estimator, SimSHAP-Semi, which is described only very briefly in Section 2.4. I don’t understand why they use this as a target rather than unbiased KernelSHAP for example. Can they verify that the model works better with this prediction target, or the the SimSHAP-Semi estimator converges faster? Previous works proposing new estimators typically perform this type of experiment (e.g., see “A Multilinear Sampling Algorithm to Estimate Shapley Values” or “Sampling Permutations for Shapley Value Estimation”) The text mentions “combining advantages of semivalue and least squares value,” but the paper doesn't actually test if this combination helps.
> > > >
> > > > Amortized explainers:
> > > > - In the description of FastSHAP following Definition 3, I still can’t tell what the authors are showing. The last response said they use $W$ that omits the $\infty$ entries for the empty and full sets, which means that the RHS of eq. 13 is *not* FastSHAP, unless we assume $g$ is constrained to make predictions to satisfy the efficiency constraint. Furthermore, the labels in the Mahalanobis loss (LHS of eq. 13) are *not* the exact Shapley values, unless we let $W$ contain the $\infty$ entries. I’m not sure what the authors’ motivation is for omitting the $\infty$ entries from $W$ in their analysis, but it seems like it invalidates this result? It might be cleaner to verify that the analysis holds when these entries are included.
> > > > - Regarding the result the authors try to show in eq. 13, it might make more sense to claim the result initially for exact Shapley values, and then separately claim it’s okay to use an approximation. And not just any approximation, but specifically one that has zero-mean noise added to the exact values (the text doesn’t mention this requirement).

---

> > > > > ### Comment · Reviewer_UWrD · 2023-12-01
> > > > > **Response (2/2)**
> > > > >
> > > > > Experiments:
> > > > > - I’m not sure why it’s necessary to normalize the curves to lie in [0, 1], is this standard for insertion/deletion? It seems to heuristically put more weight on certain examples (e.g., those that aren’t confidently classified), which doesn’t seem useful. It’s also strange that the previous results were not favorable to SimSHAP but in the odd normalized version they are. Between the normalized version and the previous version (which was in units of probability), I think the latter is preferable.
> > > > > - The y-axis isn’t labeled in Figure 4, and the caption is wrong: it shows curves, not the mean AUC.
> > > > >
> > > > > Other small things:
> > > > > - $1^S \cdot x$ looks like an inner product (dot product), whereas the authors seem to mean a Hadamard (or element-wise) product $1^S \odot x$
> > > > > - In appendix A.4, why would the authors specify that $n$ is input-specific and then write that it’s independent from $x$ with $p(x, n) = p(x)p(n)$? Those are contradictory assumptions. And why not write out what $C$ is in terms of $n$’s covariance? Also, it’s maybe better to use a symbol other than $C$, because $C$ was used in a previous equation to denote a constant we can ignore.

---

> > > > > > ### Author Response · Authors · 2023-12-01
> > > > > > **Response to Reviewer UWrD**
> > > > > >
> > > > > > We thank reviewer UWrD for another round of suggestions for our work. Due to time limitation, we have only managed to solve some minor issues and have marked them in **green** in the PDF.
> > > > > >
> > > > > > **Prior Works:**
> > > > > >
> > > > > > - **About adding the comparison to Appendix:** We also regard this as an important issue, but due to the page limit, we haven't figured out how to rearrange the structure of the text to incorporate this portion into the main body. We will revise it as part of our future work.
> > > > > > - **About difference with CoRTX：** We agree with the reviewer that CoRTX's fine-tuning stage uses MSE with ground truth labels. However, we believe that the core contribution of CoRTX is applying contrastive learning to explanation learning. Supervised RTX is similar to SimSHAP, but it also uses ground truth labels for training. However, SimSHAP does not require any labels, just like FastSHAP, and we have incorporated this aspect into the paper.
> > > > > >
> > > > > >
> > > > > >
> > > > > > **Amortized Explainers:**
> > > > > >
> > > > > > - **About Eq. 13:** The derivation in Eq.13 is the fitting target of FastSHAP (with $W$ omitting the $\infty$ entries), not the FastSHAP itself. Note that after Eq.13, we have rectified the result by normalization, which considers the $\infty$ entries.
> > > > > >
> > > > > > **Experiments:**
> > > > > >
> > > > > > - **About normalization:** We acknowledge that normalization to [0,1] is not necessary, so we revised it in the previous version of the paper to only ensure that the output for the **original image (grand coalition)** is 1. Additionally, it is worth noting that the AUC computation experimental setting has never changed, which only ensures that the output for the **original image (grand coalition)** is 1. Our previous response only aligned the curve experimental setting to be the same as the AUC computation.
> > > > > > - **About the caption and label:** Sorry for our negligence. We have revised it in the updated version of our paper.
> > > > > >
> > > > > > **Small things:**
> > > > > >
> > > > > > - **About the notation of $1^S\cdot x$:** We acknoledge that the reviewer is right, and we have revised it in the updated version.
> > > > > > - **About input-specific:**  Thanks to the reviewers for the careful review. This is a typo, and we have corrected "input-specific" to "input-agnostic".
> > > > > >
> > > > > > ## Postscript:
> > > > > >
> > > > > > Thanks to the reviewer for the detailed suggestions. However, due to the limited time at this stage, we can only complete a response to some of the issues raised in the third round of review. We will make our best effort to address all issues in the revised version of the paper that follows.

---

### Official Review · Reviewer_MdhR · 2023-11-01

**Soundness:** 3 good
**Presentation:** 3 good
**Contribution:** 3 good
**Rating:** 6
**Confidence:** 4

**Summary:**

This paper studies the efficient approximation of Shapley coefficients in the context of local feature importance for machine learning predictors. After reviewing the different approaches that approximate these intractable coefficients (exponential in dimension), the authors propose a generalization of previous approaches and a different way of computing them (employing the "LS value" distribution in the context of what they define as unified amortized estimators. They compare their results in approximation quality for tabular data, and for deletion and insertion analysis for image data, showing reasonable results, whereas the running time at inference is order of magnitude faster.

**Strengths:**

* This paper studies an important problem, with the increasing popularity of Shapley values to explain machine learning predictors (with high dimensional features).

* The paper provides a nice introduction of current approximation techniques, and their proposed variations (while not radically innovative) are simple and intuitive.

* The numerical results support their method, and seems adequate.

**Weaknesses:**

* Some typos and unclear passages makes the paper harder to read at times (see below).

* The demonstrations of benefits are all empirical, and there is no guarantee that their proposed SimSHAP provides faster approximations than other alternatives.

**Questions:**

* Why did the authors define $f:\mathcal X \to \mathcal Y$ in Sect 2.1 if this is not used in the rest of the paper? Note that the fact that $f(x_S) = v(S)$ was never made.

* The authors do a good job at commenting on related works for approximation of Shapley values, due to their complexity. However, there is no mentioning to works that provide tractable, exact or finite-sample approximate, computations of these values in cases where distributional assumptions are made:

i) [Chen, Jianbo, et al. "L-shapley and c-shapley: Efficient model interpretation for structured data." arXiv preprint arXiv:1808.02610 (2018).]

ii) [Teneggi, Jacopo et al. "Fast hierarchical games for image explanations." IEEE Transactions on Pattern Analysis and Machine Intelligence 45.4 (2022): 4494-4503]

The authors should comment on these because, in these cases, approximations are not needed, and thus Shapley coefficients admit provably more efficient computations. Naturally, if these properties of the data are not met, the approximation strategies that the authors describe do provide useful estimation tools.

* In sec 2.3, the authors first define their Unified Stochastic Estimator and then show that it is a generalization of other schemes. However, their Definition 2 reads "Most existing stochastic estimators can be unified.." which sounds like a claim to be proven, not a definition. Indeed, this is shown/proven immediately after in the form of running text. I think the authors should consider re-organizing this Definition + Remarks as Definition + Theorem, which states the strict generalization.

* I find the AUC results reported in table 3 a bit unsatisfying: the standard error (i presume?) are at the same or larger order of magnitude than the means! This makes these comparisons have very little meaning.

* On the other hand, the improvement in inference speed is massive, and the authors wait until the end, in section 4.2.4, to showcase this. This is the greatest benefit of the method, and the authors should consider stressing this throughout the text more.

* I'm confused as to how the ground truth Shapley values are computed for the experiments in Fig 2: some of the datasets have up to 96 features, which would results in completely prohibitory computation (around 10^28 flops). Even if these are computed w.r.t. some reference surrogate approximation (as I believe the authors explain in the Implementation Details), how are these surrogate trained? don't they need the ground truth values?

Minor:
* I think that to say that "Deep learning techniques have revolutionized various industries due to their learning capability by universal approximation theorem" is a stretch. Many methods provide universal approximation. The reasons for deep learning becoming so popular are others and more diverse.

* After Definition 1, the authors write "*However, machine learning models are not cooperative games and the complexity of Eq. (1) grows exponentially with the data dimension d*". How is "*machine learning models are not cooperative games*" relevant to "*the complexity of Eq. (1) grows exponentially with the data dimension d*" exactly?

* "the optimization object L in Eq. (5) need[s] to be approximated"

* There is an $\arg\min_\eta$ missing in the right-most term in Eq(5),

* capaicity -> capacity

* in equation 14, remove " \to Shapley values" as this reads as "tends to", which is not what the authors meant pressumably.

---

> ### Author Response · Authors · 2023-11-18
> **Response to Reviewer MdhR**
>
> We thank reviewer MdhR for the affirmation and pertinent suggestions on our work.
> In response to the issues raised by the reviewer, we have made the following revisions and additions.
> For the revisions of the paper, we have marked them in **green** in the PDF.
>
> ---
> ## For weaknesses
>
> Thanks for all the weaknesses pointed out, which greatly helped us revise our work.
>
> **W1:**
>
> We apologize for any confusion caused. We have carefully proofread and modified the text based on the reviewer's feedback.
>
> **W2:**
>
> It is crucial to highlight that SimSHAP belongs to the class of amortized Shapley Value estimators, and amortized techniques generally exhibit faster speeds than their non-amortized counterparts. Compared to FastSHAP, SimSHAP can achieve a **comparable** inference speed, without the need for normalization during inference.
>
> Furthermore, this article places greater emphasis on proposing a unified framework for computing Shapley values, and in the future, we intend to further explore the acceleration issue of amortized estimators from a theoretical standpoint.
>
> ---
> ## For major questions
>
> We really appreciate the reviewer's thorough examination of our work and all the insightful questions that the reviewer has raised. We have made the following adjustments to address the concerns:
>
> **Q1:**
>
> We recognize that there is room for improvement in our notations. We have streamlined the notation and removed any potential confusion.
>
> **Q2:**
>
> Thank you for recommending additional related literature. We have incorporated more references on model-specific Shapley value estimation into our revised manuscript.
>
> **Q3:**
>
> We agree that the previous presentation of **Definition 2** could be confusing. We have restructured this definition as:
> > **Unified Stochastic Estimator** is defined as the linear transformation of the values obtained from sampled subsets $S$.
>
> **Q4:**
>
> You're right that the high standard deviation values in our experiments might be misleading.
> To address this concern, we re-examined the calculation methods for metrics and implemented the bootstrap strategy recommended by FastSHAP.
> As a result, we observed a significant reduction in the standard deviation, which is clearly reflected in Table 3 of our updated version.
>
> **Q5:**
>
> Thank you for highlighting the importance of our inference speed advantage. We have emphasized this aspect in multiple sections of the "Experiments" section.
>
> **Q6:**
>
> Ground Truth Shapley value computation is not fully precise in our experiment.
> We used the `shapreg` package[^1] for the ground truth computation, which allows users to specify an error bound and compute a satisfactory ground truth Shapley value within a reasonable time span.
>
> ---
> ## For minor questions
>
> We are grateful for the minor suggestions by the reviewer in terms of writing.
>
> **Q1:**
>
> The reviewer raises an important point in this statement. We have revised the Introduction section to more clearly highlight the profound impact of deep learning techniques on various industries, due to their remarkable ability to learn complex functions both quickly and accurately.
>
> **Q2:**
>
> We acknowledge that the use of "and" in this sentence may cause some confusion, as it does not intend to establish a connection between the two sub-sentences.
> What we are trying to convey is that cooperative games do not involve the choice of value functions, whereas machine learning models are not considered cooperative games, necessitating the consideration of value function choices.
> Additionally, the complexity of computing Shapley values grows exponentially with the data dimension $d$, thus requiring the exploration of efficient estimation strategies. We have revised this portion of the text in the latest version of our paper to clarify these points.
>
> **Q3-6:**
>
> We appreciate the reviewer's careful reading. The suggestions for details are helpful and we have modified our paper based on them.
>
> ### Reference:
> [1] https://github.com/iancovert/shapley-regression

---

> ### Comment · Reviewer_MdhR · 2023-11-21
> **Responses**
>
> I thank the authors for their responses, which have clarified my doubts and comments. Before I recommend publication, and after having read all of the reviews and your responses, I would like to see the authors addressing the points raised by reviewer UWrD. In particular, I see as most important clarifying the precise similarities and differences with the references pointed out (Schwarzenberg et al and Chuang et al): if there are novel aspects relative to these works, the authors should explain what these are in detail.
>
> I see most of the other comments as minor, and I'm confident that the authors can address them (but I would like to see a plan forward on how the authors plan on doing this).

---

> > ### Comment · Reviewer_MdhR · 2023-11-22
> > **Follow up**
> >
> > As a small comment: the $\tilde{\mathcal O}(\sqrt{1/M})$ rate does indeed not follow from the law of large number, but from concentration (if $v(\cdot)$ is bounded).

---

> > > ### Author Response · Authors · 2023-11-30
> > > **Look Forward to further Comments and Suggestions**
> > >
> > > Thank you for your valuable feedback. We have carefully considered and addressed the concerns raised by other reviewers as well. If you believe our revisions address the proposed issues, would you kindly consider increasing the score? Your support is greatly appreciated!

---

### Official Review · Reviewer_Gvfa · 2023-11-21

**Soundness:** 2 fair
**Presentation:** 3 good
**Contribution:** 3 good
**Rating:** 6
**Confidence:** 3

**Summary:**

The paper presents three main contributions. First, the paper shows that existing stochastic estimators for Shapley values (semivalue, random order value, least squares value) can all be written in the form of an affine transformation of a weighted average of the values of sampled subsets. In this unified formulation, each estimator is uniquely defined by the sampling distribution for subsets, the weights in the average, and the parameters of the affine transformation. Second, the authors show that the FastSHAP objective can be viewed as minimizing the distance to estimated Shapley values under a specific metric space, noting that one can generalize this idea to define other amortized estimators by choosing other metric spaces. Then, the authors choose a particular choice of stochastic estimator (which they call Sim-Semivalue) and metric space (Euclidean) to propose SimSHAP, a new amortized Shapley estimator.

**Strengths:**

1. The unified perspectives on stochastic Shapley estimators and objectives for amortized Shapley estimators is clarifying, to my knowledge novel, and potentially quite impactful (e.g. by stimulating further progress on proposing better estimators).
2. SimSHAP seems to reach comparable performance to FastSHAP with slightly faster inference time, given FastSHAP computes an additional normalization step that SimSHAP does not.

**Weaknesses:**

1. The proposed SimSHAP seems insufficiently motivated. Why did the authors make the specific choices for the stochastic estimator and metric space that they did?
2. It is not clear from the experiments if there are any performance benefits from SimSHAP beyond faster inference.

Other notes:
1. It would be helpful if the authors were a bit more explicit about notation, e.g. explicitly defining $\mathbf{J}$ in the definition of the least squares value transformation $\mathbf{T}$ and $\mathbf{1}^{\mathbf{S}}$ in the definition of the least squares value.
2. The insertion curve seems to have two green curves but no blue curve. There might be a plotting bug somewhere?

**Questions:**

1. Why did the authors design SimSHAP the way they did?
2. What are the reasons for using SimSHAP over FastSHAP? Is the primary benefit of SimSHAP faster inference time, or are there other benefits? Also, can the authors explain why SimSHAP outperforms FastSHAP specifically in the deletion AUC for CIFAR-10 and if there is a generalizable takeaway there?
3. What do the authors mean by "That’s mostly because of the requirement of number of mask is larger for SimSHAP" when explaining why SimSHAP training is slower than FastSHAP training in Section 4.2.4 (speed evaluation)?

---

> ### Author Response · Authors · 2023-11-30
> **Response to Reviewer Gvfa**
>
> We are grateful for reviewer Gvfa for the acknowledgement and insightful suggestions. We have made targeted revisions and additions. For the revisions of the paper, we have marked them in **green** in the PDF.
>
> **Weakness 1 & Question 1:**
>
> We admit that the proposed SimSHAP is indeed theoretically
> motivated. Our motivation is that based on the unified framework, we can choose the semi-value equation and identity matrix for their simplicity. In addition, as part of our future work, we aim to identify the optimal metric matrix that maximizes performance in terms of optimization convergence.
>
> **Weakness 2 & Question 2:**
>
> - **About highlighting our contribution:** In this context, we wish to **highlight our primary contribution of unification**: Numerous recent studies have investigated various estimation techniques for approximating Shapley values, such as applying semi-value (conventional Monte Carlo estimation) and least squares value (utilizing the weighted least squares property of Shapley values) equations. We discovered that current Shapley values estimation approaches are **essentially not different** and we found that the least squares value estimation is fundamentally still equivalent to a specific type of probability Monte Carlo estimator. As a result, we put forward a unified viewpoint of stochastic estimators and built a unified perspective of amortized estimators based on this concept.
> - **About the choice between SimSHAP and FastSHAP:** With the unified framework, we proposed SimSHAP as a new perspective for Shapley value estimation, indicating its **comparable** performance to FastSHAP.
> - **About the deletion curve:** With respect to the deletion curve in our updated version of the paper, the advantage of SimSHAP is not particularly pronounced. Nevertheless, we will continue to conduct thorough analysis of the outcomes as part of our future work.
>
> **Other notes 1:**
>
> We are grateful to the reviewer for their careful reading and thoughtful suggestions regarding our notation. In our revised manuscript, we have included the definition of $J$ for clarity. Additionally, we have double-checked the definitions of $T$ and $1^S$ at the point where they first appear in the text to ensure consistency and clarity.
>
> **Other notes 2:**
>
> We have meticulously reviewed our code and experimental design. These revisions have been incorporated into the updated version of our paper.
>
> **Question 3:**
>
> We aim to demonstrate that we have recognized that SimSHAP requires a relatively large number of samples during the training process. As part of our future research, we intend to further investigate this phenomenon both theoretically and experimentally, which we believe may be related to the choice of the metric matrix.

---

> ### Author Response · Authors · 2023-11-30
> **Look Forward to further Comments and Suggestions**
>
> We would like to express our heartfelt gratitude for insightful and valuable feedback from the reviewer Gvfa. We have carefully considered each of your concerns and have implemented targeted adjustments to the original content. We are more than willing to receive further comments and suggestions from you as we continue to improve our work and responses.
>
> Should you believe our revisions have successfully addressed the issues raised, we kindly request you to adjust the scores accordingly. Your support is greatly appreciated!

---

### Meta-Review · Area_Chair_Yi9R · 2023-12-12

**Metareview:**

The paper presents a centralized view for efficient estimation of Shapley values and use this unification to develop an new estimator for Shapley values. The reviewers were mixed in their opinion with 2 shorter borderline positive reviews and one longer negative review (reviewer UWrD) where there was engagement with the authors. This negative reviewer's issues in related work and the motivation given that related work standout and when raised to the other more positive reviewers they did not disagree that the paper needs more work to integrate the related work.

**Justification For Why Not Higher Score:**

Related work and motivation given the related work is one of the most important pieces of a paper. This part of the paper needs to improve.

**Justification For Why Not Lower Score:**

N/A

---

### Decision · Program_Chairs · 2024-01-16

Reject